

# A Pan-Amazonian species delimitation: high species diversity within the genus *Amazophrynella* (Anura: Bufonidae)

Rommel R. Rojas[1], Antoine Fouquet[2], Santiago R. Ron[3], Emil José Hernández-Ruz[4], Paulo R. Melo-Sampaio[5], Juan C. Chaparro[6,7], Richard C. Vogt[8], Vinicius Tadeu de Carvalho[1,9], Leandra Cardoso Pinheiro[10], Robson W. Avila[9], Izeni Pires Farias[1], Marcelo Gordo[11] and Tomas Hrbek[1]

[1] Laboratory of Evolution and Animal Genetics, Department of Genetics, ICB, Universidade Federal do Amazonas, Brazil
[2] Laboratoire Ecologie, Evolution et Interactions des Systèmes Amazoniens, Centre de recherche de Montabo, Cayenne, French Guiana
[3] Museo de Zoología, Escuela de Biología, Pontificia Universidad Católica del Ecuador, Quito, Ecuador
[4] Laboratório de Zoologia, Faculdade de Ciências Biológicas, Campus Universitário de Altamira, Universidade Federal do Pará, Altamira, Para, Brazil
[5] Departamento de Vertebrados, Museu Nacional, Rio de Janeiro, Rio de Janeiro, Brazil
[6] Coleccion de anfibios y reptiles, Museo de la Biodiversidad, Cusco, Peru
[7] Museo de Historia Natural, Universidad Nacional de San Antonio Abad, Cusco, Peru
[8] CEQUA, Coordenação de Biodiversidade, Instituto Nacional de Pesquisas da Amazônia, Manaus, Amazonas, Brazil
[9] Departamento de Ciências Biológicas, Centro de Ciências Biológicas e da Saúde, Universidade Regional do Cariri, Crato, Ceara, Brazil
[10] Museu Paraense Emilio Goeldi, Belem, Para, Brazil
[11] Departamento de Biologia, ICB, Universidade Federal do Amazonas, Manaus, Amazonas, Brazil

Corresponding authors
Rommel R. Rojas,
rrojaszamora@gmail.com
Tomas Hrbek, hrbek@evoamazon.net

## ABSTRACT

Amphibians are probably the most vulnerable group to climate change and climate-change associate diseases. This ongoing biodiversity crisis makes it thus imperative to improve the taxonomy of anurans in biodiverse but understudied areas such as Amazonia. In this study, we applied robust integrative taxonomic methods combining genetic (mitochondrial 16S, 12S and COI genes), morphological and environmental data to delimit species of the genus *Amazophrynella* (Anura: Bufonidae) sampled from throughout their pan-Amazonian distribution. Our study confirms the hypothesis that the species diversity of the genus is grossly underestimated. Our analyses suggest the existence of eighteen linages of which seven are nominal species, three Deep Conspecific Lineages, one Unconfirmed Candidate Species, three Uncategorized Lineages, and four Confirmed Candidate Species and described herein. We also propose a phylogenetic hypothesis for the genus and discuss its implications for historical biogeography of this Amazonian group.

## INTRODUCTION

Amphibians are undergoing a drastic global decline (*Beebee & Griffiths, 2005*). This decline is primarily attributable to habitat destruction, diseases (chytrid fungus) and global climate change (*Collins, 2010*). In Amazonia the primary threat is habitat destruction, although the chytrid fungus has reached the Amazon basin (*Valencia-Aguilar et al., 2015*; *Becker et al., 2016*), and is starting to have an impact on Amazonian and Andean anurans (*Lötters et al., 2005*; *Lötters et al., 2009*; *Catenazzi & Von May, 2014*). Most Amazonian amphibians are thought to have broad, often basin wide distributions, although their geographic distributions are generally poorly known. More detailed analyses generally reveal the existence of multiple deeply divergent lineages, suggesting cryptic diversity. *Fouquet et al. (2007a)* and *Fouquet et al. (2007b)* estimated that amphibian diversity of Amazonia is underestimated by 115%, while *Funk, Caminer & Ron (2012)* suggest this underestimate is closer to 150–350%. But even without taking into account the high levels of crypsis or pseudocrypsis (morphological differences apparent but overlooked) in widespread Amazonian anurans, Amazonia has the highest diversity of amphibians on this planet (*Jenkins, Pimm & Joppa, 2013*).

Delimiting species and their geographic distributions is therefore crucial for the understanding of impacts on the biodiversity of Amazonian anurans, and for the assessment of their conservation status (*Angulo & Reichle, 2008*). Previous studies suggest a prevalent conservatism in the morphological evolution of anurans (e.g., *Elmer, Dávila & Lougheed, 2007*; *Robertson & Zamudio, 2009*; *Vences et al., 2010*; *Kaefer et al., 2012*; *Rowley et al., 2015*), thus, species delimitation based solely on morphological characters may fail to differentiate among species. Conversely, delimiting species solely based on molecular characters or genetic distances harbors potential pitfalls that have been well documented (e.g., *Carstens et al., 2013*; *Sukumaran & Knowles, 2017*). Environmental data also have the potential to provide important information to taxonomy since species have distinct ecological requirements that determine their occurrence in time and space (*Soberón & Peterson, 2005*). Therefore, species delimitation relying on a pluralistic approach seeking to unite several lines of evidence (*Dayrat, 2005*; *Padial et al., 2010*) generally provides robust and consensual taxonomic hypotheses (e.g., *Padial & De La Riva, 2009*) especially in morphologically conserved groups, i.e., taxonomic groups harboring cryptic or pseudocryptic taxa (*Cornils & Held, 2014*).

The frog genus *Amazophrynella Fouquet et al., 2012a* is distributed throughout Amazonia, and currently comprises seven small-sized (12.0–25.0 mm) species (*Fouquet et al., 2012b*). All species inhabit the forest leaf litter (*Rojas et al., 2015*), breed in seasonal pools and have diurnal and crepuscular habits (*Fouquet et al., 2012b*; *Rojas et al., 2014*; *Rojas et al., 2016*).

Until 2012, only two species were recognized: *Amazophrynella minuta* from western Amazon and *Amazophrynella bokermanni* from eastern Amazon (*Fouquet et al., 2012b*). Since 2012 five additional species have been described from western Amazon (*Amazophrynella vote*, *Amazophrynella manaos*, *Amazophrynella amazonicola*, *Amazophrynella matses* and *Amazophrynella javierbustamantei*). The taxonomy of the genus

remains, however, far from being resolved (*Rojas et al., 2016*). Although molecular phylogenetic analyses in *Fouquet et al. (2012b)*, *Rojas et al. (2015)* and *Rojas et al. (2016)* provided evidence for the existence of multiple lineages, the scarcity of material suitable for morphological and bioacoustic analyses prevented the description of these lineages as new species.

In this study, we revisit the genus *Amazophrynella*, include specimens from new localities, and reconstruct intra- and inter-specific phylogenetic relationships. We delimit candidate species based on molecular data and subsequently seek support for these lineages combining qualitative and quantitative morphological data and environmental evidence. As a result of these analyses, we formally describe four new species of *Amazophrynella* from Brazil, Ecuador, French Guiana and Peru, and identify additional seven candidate species. Additionally, we provide new insights into the overall phylogenetic relationships for the genus, and discuss biogeographic history of this Amazonian group.

## MATERIAL AND METHODS

### Protocol for species delimitation

We evaluated the status of populations of *Amazophrynella*, adhering to the unified species concept proposed by *Queiroz De (2007)*, that conceptualizes species as lineages of ancestor-descendent populations which maintain their distinctness from other such lineages and which have their own evolutionary tendencies and historical fates. We followed the consensus protocol of integrative taxonomy proposed by *Padial et al. (2010)*. The concept of candidate species adopted in this study follows the subcategories proposed by *Vieites et al. (2009)* in using: Confirmed Candidate Species (CCS) for lineages that present high genetic distance and can be differentiated by other traits (i.e., morphological data), Deep Conspecific Lineages (DCL) for lineages that are genetically divergent but not supported by any other character (these characters being available), Unconfirmed Candidate Species (UCS) for lineages that are genetically divergent but no additional characters are available to support this divergence (these characters not available) and Uncategorized Lineages (UL) for lineages that do not corresponds to any of the above categories.

### Focal species and morphological examination

Field work and visits to museum collections were carried out between 2011 and 2017. Field collection of specimens followed the technique of visual encounter surveys and pitfall-barrier traps (*Crump & Scott Jr, 1994*). Museum acronyms are found in *Sabaj (2016)* except for Museo de Biodiversidad del Peru (MUBI; this collection is part of Museo de Historia Natural, Universidad Nacional de San Antonio Abad, Cusco, Peru). Collecting permits in Peru were granted by Dirección General Forestal y de Fauna Silvestre del Ministerio del Medio Ambiente (MINAN; No. AUT-IFS-2017-055), in Ecuador by Ministerio del Ambiente (MA; 001-1-IC-FAU-DNB/MA) and in Brazil by the Instituto Chico Mendes de Conservação da Biodiversidade (ICMBio; No. 39792-1 and No. 32401). The material of *Amazophrynella teko* from Mitaraka (French Guiana) was collected during the "Our Planet Reviewed" expedition, organized by the MNHN and Pro-Natura International.

We examined topotypical material of *A. minuta* deposited at the collection of Amphibians and Reptiles of the Instituto Nacional de Pesquisas da Amazônia–INPA

(INPA–H) and three syntypes (NHMG 462, NHMG 463, NHMG 464) deposited at the Göteborgs Naturhistoriska Museum, Sweden; five specimens of *A. bokermanni* (*Izecksohn, 1993*) from near the type locality (*c.* 30 Km) deposited at the INPA collection; the type series of *A. vote* (*Ávila et al., 2012*) deposited at the Coleção Zoológica de Vertebrados of the Universidade Federal de Mato Grosso–UFMT, Cuiabá, Mato Grosso, Brazil (UFMT–A) and INPA; *A. manaos* (*Rojas et al., 2014*) deposited at the INPA; *A. amazonicola* and *A. matses* (*Rojas et al., 2015*) deposited at the Museo de Zoología–Universidad Nacional de la Amazonia Peruana–UNAP and *A. javierbustamantei* (*Rojas et al., 2016*) deposited at the Museo de Biodiversidad del Peru (MUBI), Museo de Historia Natural de la Universidad Nacional Mayor de San Marcos (MHNSM). A list of the examined specimens is found in Appendix S1.

Qualitative morphological terminology was according to *Kok & Kalamandeen (2008)*. Morphological comparison between specimens were made through visual inspection of diagnostic characters that include: dorsal skin texture, ventral skin texture, head shape, shape of palmar tubercle, relative length of fingers and venter coloration (*Fouquet et al., 2012b*; *Rojas et al., 2014*; *Rojas et al., 2015*; *Rojas et al., 2016*). We used ventral incision to perform gonadal analyses. Developmental stages of tadpoles were determined using Gosner's protocol (*1960*). Descriptive terminology, morphometric variables and developmental stages of tadpoles follow *Altig & McDiarmid (1999)*. Spectral and temporal parameters of advertisement calls (when available) were analyzed in the software Praat for Windows (*Boersma & Weenick, 2006*). Bioacoustic terminology followed *Köhler et al. (2017)*.

## Morphological quantitative analyses

Quantitative measurements of body were obtained with a digital caliper (0.1 mm precision) following *Kok & Kalamandeen (2008)* with the aid of an ocular micrometer in a Leica stereomicroscope. Measurements were taken from the right side of specimens, and, if this was not feasible, from the left side. Measurements were: SVL (snout-vent length) from the tip of the snout to the posterior margin of the vent; HL (head length) from the posterior edge of the jaw to the tip of the snout; HW (head width), the greatest width of the head, usually at the level of the posterior edges of the tympanum; ED (eye diameter); IND (internarial distance), the distance between the edges of the nares; SL (snout length) from the anterior edge of the eye to the tip of the snout; HAL (hand length) from the proximal edge of the palmar tubercle to the tip of finger III; UAL (upper arm length) from the edge of the body insertion to the tip of the elbow; THL (thigh length) from the vent to the posterior edge of the knee; TL (tibia length) from the outer edge of the knee to the tip of the heel; TAL (tarsal length) from the heel to the proximal edge of the inner metatarsal tubercle; FL (foot length) from the proximal edge of the inner metatarsal tubercle to the tip of toe IV. We rounded all measurements to first decimal place to avoid pseudoprecision (*Hayek, Heyer & Gascon, 2001*).

Principal Component Analyses (PCA) were performed on residuals obtained by linear-regressing each variable on SVL, thus removing the effects of size. We used only males specimens because of absence of females in some lineages. The PCA was used to detect groups representing putative species. We also performed a discriminant Function

Analysis (DFA) to identify morphometric variables that contribute the most to species separation and to test the classification of specimens into mtDNA lineages. For DFA we used morphometric size-free data set. To determine the number of correct and incorrect assignments of specimens to each of the mtDNA lineages, we jackknifed our data matrix. The significance of differences of morphological variables among mtDNA lineages was tested using the Kruskal–Wallis (KW) non-parametric test. All the statistical analyses (PCA, DFA and KW) were performed in R v3.4.3 (*R Development Core Team, 2008*) using the stats package and setting the significance cut–off at 5%.

## DNA amplification

DNA extraction, gene amplification and sequencing was carried out using standard protocols (Appendix S2). Sequence data were deposited in GenBank under the accession numbers MH269714–MH270330 (Table S2A).

## Phylogenetic analyses and species delimitation

We collected molecular data for 230 individuals of *Amazophrynella* from 35 localities, including topotypical material for all nominal species and encompassing the entire distribution of the genus. We obtained a total of 1,430 bp from three mitochondrial loci (16S rRNA (16S), 480 bp; 12S rRNA (12S), 350 bp; and Cytochrome oxidase subunit I (COI), 600 pb (see Appendix S4, Table S4A)). The edition and alignment of the sequences was performed using Geneious v.6.1.8. (*Kearse et al., 2012*) and the Clustal W algorithm (*Thompson, Gibson & Higgins, 2002*). We used only unique haplotypes for phylogenetic reconstruction. We concatenated all loci, treating them as a single partition evolving under the same model of molecular evolution. The best model of molecular evolution (GTR+G+I) was estimated in JModelTest (*Posada, 2008*) and selected using the Akaike Information Criterion–AIC. Phylogenetic analyses were performed using Bayesian Inference (BI) using MrBayes 3.2.1. (*Huelsenbeck & Ronquist, 2001*). We generated $10^7$ topologies, sampling every 1,000 th topology and discarding the first 10% topologies as burn-in. The stationarity of the posterior distributions for all model parameters was verified in Tracer v1.5 (*Rambaut & Drummond, 2009*). From the MCMC output, we generated the final consensus tree-maximum clade credibility tree using Tree Annotator v1.6.2 (part of the Beast software package). For visualization and edition of the consensus maximum clade credibility tree, we used the program Figtree v.1.3. (*Rambaut, 2009*).

We used a Poisson tree processes (PTP) model (*Zhang et al., 2013*) to infer the most likely number of species in our dataset, as implemented in the bPTP server (http://species.h-its.org/ptp/). The PTP model is a simple, fast and robust algorithm to delimit species using non-ultrametric phylogenies, ultrametricity is not required because the algorithm models speciation rates by directly using the number of substitutions. The fundamental assumption is that the number of substitutions between species is significantly higher than the number of substitutions within species. In a sense, this is analogous to the GMYC (General Mixed Yule Coalescent) approach that seeks to identify significant changes in the rate of branching events on the tree. However, GMYC uses time to identify branching rate transition points, whereas, in contrast, PTP directly uses the number of

substitutions (*Zhang et al., 2013*). For input, we used a BI tree estimated by MrBayes. We ran the PTP analyses using $10^5$ MCMC generations, thinning value of 100, a burn-in of 10%, and opted for remove the outgroup to improve species delimitation. Convergence of MCMC chain was confirmed visually. To ensure that the lineages detected using PTP presented high genetic distance (>3.0%, *sensu* (*Fouquet et al., 2007a*; *Fouquet et al., 2007b*) we calculated uncorrected *p*-distance using the 16S mtDNA (*Vences et al., 2005*) in the program MEGA 7.0 (*Kumar, Stecher & Tamura, 2016*).

To generate a dated tree in Beast 2.0 (*Drummond & Rambaut, 2007*), we selected one representative individual per species. We used a birth and death prior, GTR+I+G evolution model and calibrated the tree using normal distribution following the divergence time estimates of *Fouquet et al. (2012a)*: crown age of Hyloidea (mean = 77.0 ± 10 Ma); basal divergence time of Bufonidae (mean = 67.9 ± 12 Ma); divergence of *Atelopus + Oreophrynella* vs. other Bufonidae (mean = 60.0 ± 11 Ma); *Nannophryne* vs. other Bufonidae (mean = 47.0 ± 8 Ma); *Rhaebo* vs. other crown Bufonidae (mean = 40.8 ± 7 Ma) and *Dendrophryniscus* vs. other crown Bufonidae (mean = 52.1 ± 9). We generated $10^7$ topologies, sampling every 1,000 th topology and discarding the first 10% topologies as burn-in. Stationarity of the posterior distributions for all model parameters was verified in Tracer v1.5 (*Rambaut & Drummond, 2009*). From the MCMC output, we generated the final consensus maximum clade credibility tree using Tree Annotator v1.6.2 (part of Beast software package). For visualization and edition of the consensus tree, we used the program Figtree v.1.5 (*Rambaut, 2009*).

## Environmental analyses

The environmental analyses were undertaken in order to test if delimited species occur in distinct climatic environments (*Soberón & Peterson, 2005*). We retrieved high resolution bioclimatic layers (30 arc–seconds ∼1 km, present environmental conditions) using the Community Climate System Model (CCSM4) from the WorldClim project (http://www.worldclim.org/) (*Hijmans et al., 2005*). To avoid geographic pseudoocurrence of points, localities were filtered using the program Geographic Distance Matrix Generator 1.2.3. (*Ersts, 2014*) considering a threshold of 1 km between localities. The localities of each lineage used for analyses are in Appendix S3, Table S3A.

To identify environmental variables that were most informative and test the classification of specimens into mtDNA lineages using ecological variables, we performed Principal Component Analysis (PCA) and Discriminant Function Analysis (DFA) separately for each lineages/species of the eastern and western clades. The analyses were performed using the 19 BioClim environmental variables in WordClim. Probability of correct assignment of individuals to lineages was tested using jackknife.

## Electronic publication of new zoological taxonomic names

The electronic version of this article in Portable Document Format (PDF) will represent a published work according to the International Commission on Zoological Nomenclature (ICZN), and hence the new names contained in the electronic version are effectively published under that Code from the electronic edition alone. This published work

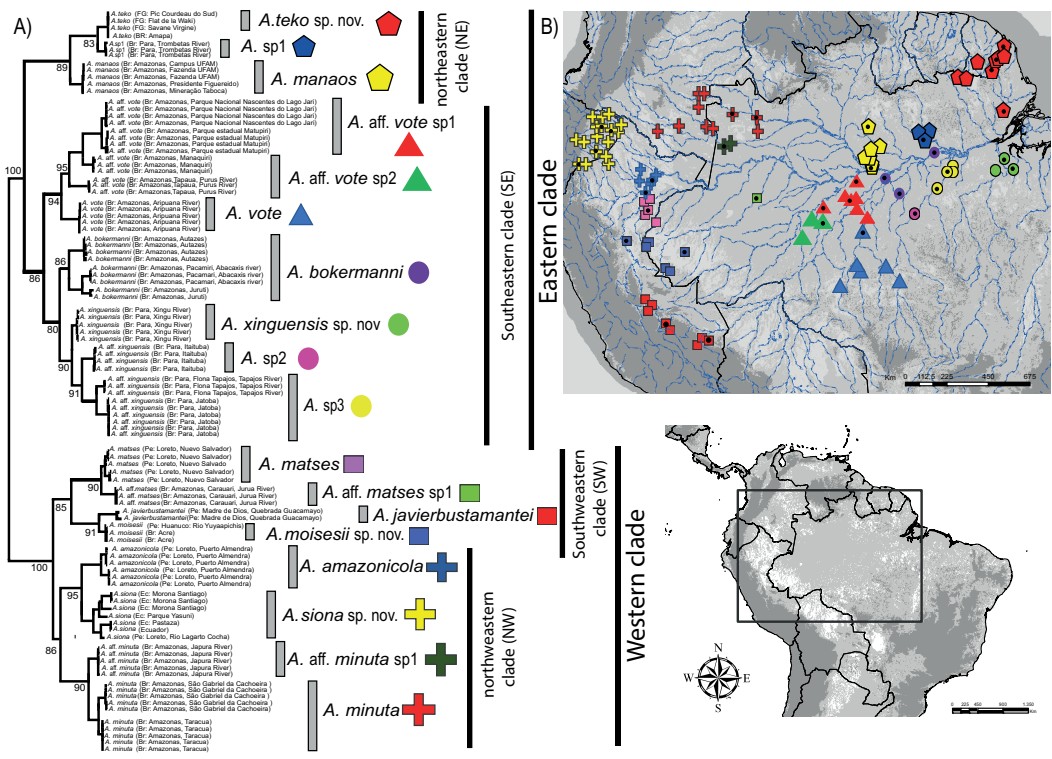

**Figure 1** **Phylogeny and geographic distribution of *Amazophrynella*.** (A) Phylogenetic relationship among nominal and putative species of *Amazophrynella* based on Bayesian inference inferred from 1,430 aligned sites of the 16S, 12S and COI mtDNA genes. Numbers in branches represent Bayesian posterior probability. (B) Geographic distribution of *Amazophrynella* spp. Colors and symbols = occurrence areas for each clade based on specimens reviewed in collections. Black points = localities of genetic collection from specimens. Colors and symbols of clades in the phylogenetic tree correspond to colors and symbols on the map.

and the nomenclatural acts it contains have been registered in ZooBank, the online registration system for the ICZN. The ZooBank LSIDs (Life Science Identifiers) can be resolved and the associated information viewed through any standard web browser by appending the LSID to the prefix http://zoobank.org/. The LSID for this publication is: urn:lsid:zoobank.org:pub:1C6046BE-CFC4-4060-A1CA-0C9C9C1C7A0A. The online version of this work is archived and available from the following digital repositories: PeerJ, PubMed Central and CLOCKSS.

# RESULTS

## Phylogenetic and species diversity

The concatenated data resulted in a strongly supported phylogeny (Fig. 1), with high degree of divergence among putative and nominal species of *Amazophrynella*. The PTP model of species delimitation detected a total of eighteen lineages (posterior probability = 0.48–0.91) (Appendix S4, Fig. S4A) of which seven are nominal species and 11 are candidate species.

**Table 1  Lineages and taxonomic status.** Uncorrected *p*–distances among mtDNA lineages of *Amazophrynella*. Molecular distances are based on the 480–bp fragment of 16S rDNA.

| | | 1 | 2 | 3 | 4 | 5 | 6 | 7 | 8 | 9 | 10 | 11 | 12 | 13 | 14 | 15 | 16 | 17 |
|---|---|---|---|---|---|---|---|---|---|---|---|---|---|---|---|---|---|---|
| 1 | *A. amazonicola* | | | | | | | | | | | | | | | | | |
| 2 | *A. siona* sp. nov. | 0.07 | | | | | | | | | | | | | | | | |
| 3 | *A.* aff. *minuta* sp1 | 0.08 | 0.09 | | | | | | | | | | | | | | | |
| 4 | *A. minuta* | 0.09 | 0.09 | 0.02 | | | | | | | | | | | | | | |
| 5 | *A. matses* | 0.09 | 0.13 | 0.09 | 0.09 | | | | | | | | | | | | | |
| 6 | *A.* aff. *matses* sp1 | 0.09 | 0.13 | 0.09 | 0.10 | 0.02 | | | | | | | | | | | | |
| 7 | *A. javierbustamantei* | 0.09 | 0.13 | 0.08 | 0.08 | 0.06 | 0.06 | | | | | | | | | | | |
| 8 | *A. moisesii* sp. nov. | 0.08 | 0.11 | 0.08 | 0.08 | 0.09 | 0.09 | 0.06 | | | | | | | | | | |
| 9 | *A. vote* | 0.12 | 0.15 | 0.11 | 0.11 | 0.13 | 0.13 | 0.11 | 0.10 | | | | | | | | | |
| 10 | *A.* aff. *vote* sp1 | 0.12 | 0.15 | 0.11 | 0.11 | 0.12 | 0.12 | 0.12 | 0.11 | 0.03 | | | | | | | | |
| 11 | *A.* aff. *vote* sp2 | 0.12 | 0.15 | 0.11 | 0.11 | 0.12 | 0.12 | 0.12 | 0.11 | 0.04 | 0.03 | | | | | | | |
| 12 | *A. bokermanni* | 0.12 | 0.14 | 0.11 | 0.11 | 0.12 | 0.12 | 0.11 | 0.11 | 0.05 | 0.05 | 0.06 | | | | | | |
| 13 | *A.* sp2 | 0.12 | 0.15 | 0.10 | 0.11 | 0.11 | 0.11 | 0.11 | 0.11 | 0.07 | 0.08 | 0.08 | 0.07 | | | | | |
| 14 | *A.* sp3 | 0.11 | 0.14 | 0.10 | 0.10 | 0.11 | 0.11 | 0.12 | 0.10 | 0.07 | 0.07 | 0.07 | 0.06 | 0.04 | | | | |
| 15 | *A. xinguensis* | 0.12 | 0.15 | 0.11 | 0.12 | 0.13 | 0.13 | 0.13 | 0.11 | 0.07 | 0.08 | 0.08 | 0.07 | 0.05 | 0.06 | | | |
| 16 | *A. manaos* | 0.13 | 0.15 | 0.12 | 0.13 | 0.11 | 0.11 | 0.12 | 0.12 | 0.09 | 0.09 | 0.08 | 0.09 | 0.09 | 0.09 | 0.09 | | |
| 17 | *A.* sp1 | 0.12 | 0.15 | 0.11 | 0.12 | 0.11 | 0.12 | 0.12 | 0.13 | 0.11 | 0.10 | 0.09 | 0.10 | 0.09 | 0.10 | 0.10 | 0.06 | |
| 18 | *A. teko* sp. nov. | 0.12 | 0.15 | 0.11 | 0.12 | 0.11 | 0.12 | 0.12 | 0.13 | 0.10 | 0.10 | 0.09 | 0.09 | 0.09 | 0.09 | 0.09 | 0.05 | 0.03 |

The phylogeny of *Amazophrynella* recovered the presence of two clades diverging basally, both strongly supported: one distributed in eastern and other in western Amazonia (see Fig. 1A). The eastern clade was formed by two strongly supported subclades, herein called northeastern (NE) and southeastern (SE) clades. The northeastern clade included three lineages and the southeastern clade seven lineages. The western clade was formed by two well supported subclades, herein called northwestern (NW) and southwestern (SW) clades. Both subclades were composed of four lineages (see Fig. 1A). Uncorrected *p*-distances for 16S mtDNA between pairs of sister lineages are presented in Table 1. Each lineage presented high genetic divergence (>3.0%) compared to its sister taxon and ranged between 3.0–3.2% (3.0 ± 0.1) to 4.0–6.0% (5.0 ± 0.1).

Our timetree recovered *Dendrophryniscus* as sister taxon of *Amazophrynella* (see Appendix S5, Fig. S5 a for complete timetree calibration), with a divergence time estimated at 38.1 Ma (95% HPD: 49.0–29.0 Ma), an Eocene divergence, with strong support (pp = 1.0, see Fig. 2). Within *Amazophrynella* the eastern/western divergence was estimated at 24.8 Ma (95% HPD: 30.0–19.0 Ma), a Late Oligocene to Early Miocene divergence. Within the eastern clade the SE and NE subclades diverged during the Early Miocene (20.1 Ma, 95% HPD: 22.0–18.0 Ma). In the western clade, the split between the NW and SW subclades was estimated at 16.5 Ma (95% HPD = 18.0–13.0 Ma), a Middle Miocene divergence. Divergence time between each pair of lineages within each of the four above clades varied between 10.8 and 2.1 Ma.

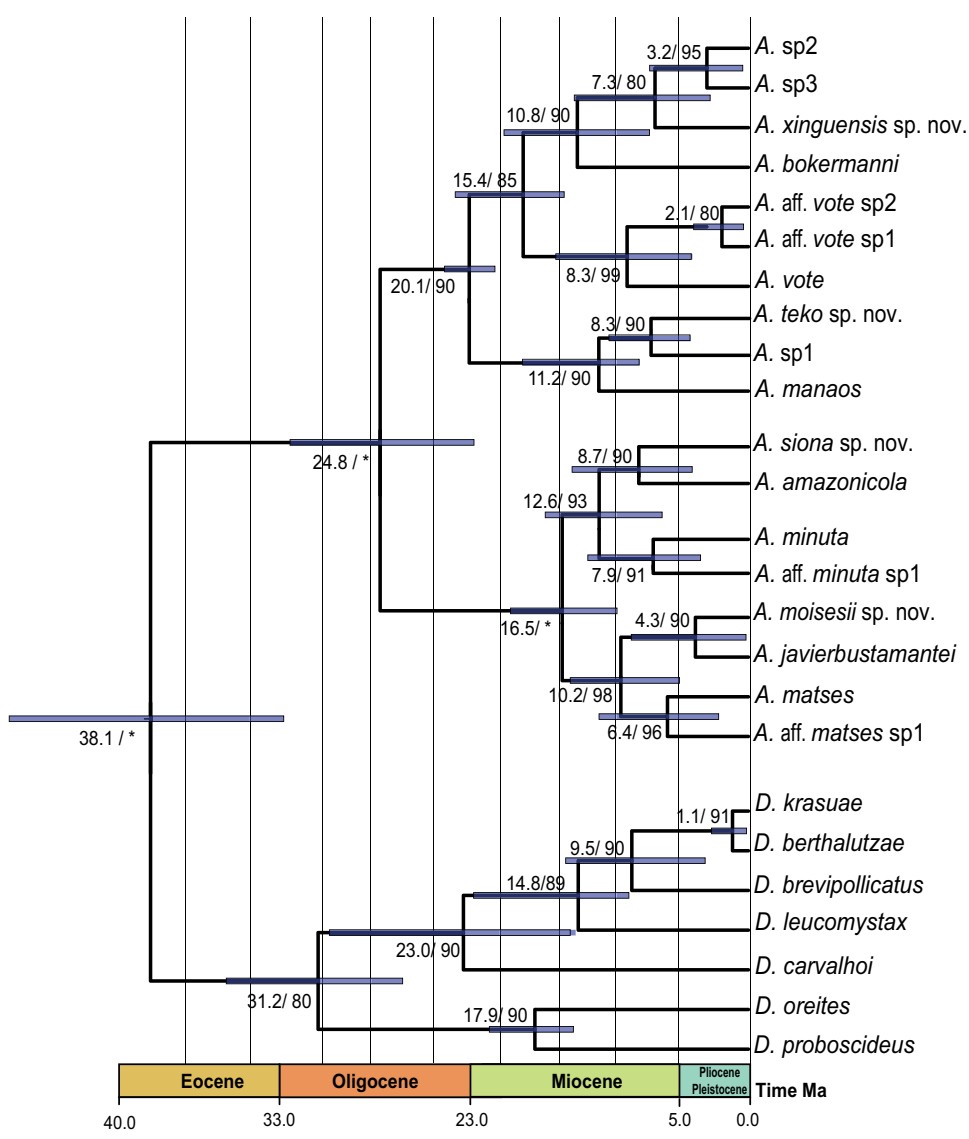

**Figure 2  Timetree of *Amazophrynella*.** Time calibrated tree of *Amazophrynella* with posterior probabilities and mean age. Blue bars represent 95% HPD.

## Morphological analyses

A total of 468 specimens (adult males and females) were examined for comparative morphological analyses (Table 2); these analyses did not include *Amazophrynella* aff. *matses* sp, *A*.sp2 and *A* sp3 (see Fig. 1). Measurements of males and females are presented in Tables 3 and 4. For morphometric analyses (Principal Components Analyses-PCA and Discriminant Function Analyses-DFA) we used 237 adult male specimens (87 from the eastern clade and 148 from the western clade). The specimens used in morphometric analyses are listed in Appendix S6.

**Table 2 Lineage classification and diagnostic characters.** Taxonomic status, congruence and comparison of main diagnostic morphological characters of species identified in phylogenetic analyses (16S + 12S + COI). Character (–) indicates no data available. CCS = Confirmed Candidate Species; UCS = Unconfirmed Candidate Species; DCL = Deep Conspecific Lineages; UL = Uncategorized Lineage.

| Lineages | Status | Dorsal skin texture | Ventral skin texture | Head shape | Palmar tubercle | FI vs. FII | Venter coloration | Venter stain |
|---|---|---|---|---|---|---|---|---|
| *A. manaos* | CCS | Granular | Granular | Truncate | Elliptical | I < II | White | Large blotches |
| *A. teko* sp. nov. | CCS | Highly granular | Highly granular | Acute | Elliptical | I < II | Creamy | Small blotches |
| *A.* sp1 | UL | Highly granular | Highly granular | Acute | Elliptical | I < II | Creamy | Small blotches |
| *A. vote* | CCS | Tuberculate | Granular | Rounded | Rounded | I < II | Reddish-brown | Small dots |
| *A.* aff. *vote* sp1 | DCL | Tuberculate | Granular | Rounded | Rounded | I < II | Reddish-brown | Small dots |
| *A.* aff. *vote* sp2 | DCL | Tuberculate | Granular | Rounded | Rounded | I < II | Reddish-brown | Small dots |
| *A. bokermanni* | CCS | Granular | Granular | Pointed | Rounded | I>II | white | Small dots |
| *A. xinguensis* sp. nov. | CCS | Highly granular | Granular | Pointed | Ovoid | I = II | Greyish | Medium-size dots |
| *A.* sp2 | UL | – | – | – | – | – | – | – |
| *A.* sp3 | UL | – | – | – | – | – | – | – |
| *A. matses* | CCS | Spiculate | Granular | Acute | Rounded | I < II | Yellow | Blotches |
| *A.* aff. *matses* sp1 | UCS | – | – | – | – | – | – | – |
| *A. javierbustamantei* | CCS | Tuberculate | Coarsely areolate | Acuminate | Rounded | I < II | Pale yellow | Small dots |
| *A. moisesii* sp. nov. | CCS | Tuberculate | Highly granular | Acuminate | Elliptical | I < II | Pale yellow | Tiny points |
| *A. amazonicola* | CCS | Finely granular | Granular | Pointed | Rounded | I < II | Yellow | Medium-size blotches |
| *A. siona* sp. nov. | CCS | Finely granular | Granular | Acute | Rounded | I < II | Reddish-brow | Small blotches |
| *A. minuta* | CCS | Highly granular | Granular | Pointed | Rounded | I < II | Yellow-orange | Large blotches |
| *A.* aff. *minuta* sp1 | DCL | Highly granular | Granular | Pointed | Rounded | I < II | Yellow-orange | Large blotches |

The PCA of the eastern and western clades revealed a grouping of specimens based on morphometric traits and allowed us to distinguish all the mtDNA lineages in multivariate space (Figs. 3A and 3B). Character loadings, eigenvalues and percentage of variance explained for PCA (PC I-II) for morphometric variables for the eastern and western clades are provided in Appendix S7 and Table S7A–S7B.

In the eastern clade specimens of each lineage can be successfully separated based on morphometric traits using PCA (Fig. 3A). The first two principal components extracted by the PCA account for 57.7% of the variation found in the dataset. The first component (PC1) explained 37.48% of the total variation and the second component (PC2) explained 20.29% of the variation. Using DFA a total of 80% of specimens were correctly classified to phylogenetic groups. The number of individuals correctly assigned to each clade by DFA are presented in Table 5. The DFA showed that the variables that contributed the most to the morphometric separation were snout length, tarsal length, and head width. Head measurement traits (head width, head length, snout length, and intranasal distance) explained 93% of the classification by the first two discriminant axes (Appendix S8, Figs. S8A–S8B). Loadings and percentage of variance explained for discriminant axes (F1–2) of morphometric variables in eastern clade are provided in Appendix S8 and Table S8A).

In the western clade specimens of each lineage can be successfully separated based on morphometric traits using PCA (Fig. 3B). The first two principal components extracted

Rojas et al. (2018), *PeerJ*, DOI 10.7717/peerj.4941

**Table 3  Male descriptive morphometric statistics.** Descriptive morphometric statistics (in mm) for males of nominal and CCE of *Amazophrynella*. KW = Kruskal Wallis test, (+) *p*-value < 0.05.

| Variable | A. minuta (n = 20) | A. matses (n = 13) | A. javierbustamantei (n = 28) | A. moisesii usp. nov (n = 15) | A. amazonicola (n = 15) | A. siona sp. nov. (n = 29) | A. bokerma-nni (n = 7) | A. xinguensis sp.nov. (n = 5) | A. manaos (n = 27) | A. teko sp.nov. (n = 13) | A. vote (n = 14) | KW p-value |
|---|---|---|---|---|---|---|---|---|---|---|---|---|
| SVL | 13.5 ± 0.6 | 12.1 ± 0.6 | 14.9 ± 0.9 | 14.3 ± 0.5 | 14.5 ± 0.7 | 13.1 ± 0.6 | 16.3 ± 0.2 | 18.8 ± 0.9 | 14.2 ± 0.7 | 14.8 ± 0.7 | 13.1 ± 0.7 | + |
| HW | 4.2 ± 0.2 | 3.6 ± 0.2 | 4.2 ± 0.2 | 4.3 ± 0.4 | 4.4 ± 0.3 | 3.9 ± 0.3 | 4.8 ± 0.1 | 5.1 ± 0.2 | 4.2 ± 0.3 | 4.5 ± 0.3 | 4.0 ± 0.7 | + |
| HL | 4.9 ± 0.2 | 4.3 ± 0.3 | 5.1 ± 0.3 | 5.4 ± 0.3 | 5.2 ± 0.3 | 4.9 ± 2.2 | 5.7 ± 0.1 | 6.6 ± 0.2 | 5.3 ± 0.3 | 5.3 ± 0.2 | 4.6 ± 0.3 | + |
| SL | 2.3 ± 0.1 | 2.0 ± 0.3 | 2.2 ± 0.2 | 2.6 ± 0.2 | 2.4 ± 0.2 | 2.2 ± 0.2 | 3.0 ± 0.1 | 3.2 ± 0.1 | 2.7 ± 0.2 | 2.5 ± 0.1 | 2.1 ± 0.2 | + |
| ED | 1.4 ± 0.1 | 1.1 ± 0.1 | 1.3 ± 0.1 | 1.6 ± 0.2 | 1.2 ± 0.1 | 1.3 ± 0.1 | 1.7 ± 0.1 | 2.0 ± 0.1 | 1.3 ± 0.1 | 1.5 ± 0.1 | 1.3±.1 | + |
| IND | 1.2 ± 0.1 | 1.0 ± 0.1 | 0.9 ± 0.1 | 1.2 ± 0.1 | 1.2 ± 0.1 | 1.1 ± 0.08 | 1.4 ± 0.1 | 1.5 ± 0.5 | 1.1 ± 0.1 | 1.3 ± 0.1 | 1.1 ± 0.1 | + |
| UAL | 3.8 ± 0.2 | 3.5 ± 0.4 | 4.5 ± 0.4 | 4.8 ± 0.6 | 4.5 ± 0.3 | 4.1 ± 0.4 | 5.4 ± 0.4 | 6.1 ± 0.5 | 3.6 ± 0.4 | 4.8 ± 3.2 | 3.9 ± 0.5 | + |
| HAL | 2.8 ± 0.2 | 2.7 ± 0.2 | 3.6 ± 0.4 | 3.4 ± 0.5 | 3.2 ± 0.2 | 2.7 ± 0.2 | 3.4 ± 0.6 | 3.7 ± 0.3 | 2.8 ± 0.6 | 3.2 ± 0.2 | 3.0 ± 0.3 | + |
| THL | 6.8 ± 0.2 | 6.2 ± 0.4 | 7.6 ± 0.7 | 7.9 ± 0.8 | 7.7 ± 0.6 | 7.0 ± 0.4 | 8.0 ± 0.3 | 9.5 ± 0.8 | 6.7 ± 0.3 | 7.6 ± 0.8 | 6.5 ± 0.7 | + |
| TAL | 6.7 ± 0.3 | 5.8 ± 0.3 | 7.6 ± 0.7 | 7.7 ± 0.9 | 7.2 ± 0.6 | 6.6 ± 0.4 | 7.5 ± 0.3 | 9.1 ± 0.7 | 6.9 ± 0.6 | 7.3 ± 0.5 | 5.7 ± 0.7 | + |
| TL | 4.1 ± 0.2 | 3.8 ± 0.2 | 4.7 ± 0.8 | 5.2 ± 1.2 | 4.2 ± 0.6 | 4.1 ± 0.4 | 4.8 ± 0.4 | 5.5 ± 0.2 | 4.6 ± 0.4 | 4.6 ± 0.4 | 3.8 ± 1.0 | + |
| FL | 4.8 ± 0.4 | 4.3 ± 0.4 | 5.7 ± 0.6 | 5.7±0.7 | 5.1 ± 0.4 | 4.7 ± 0.5 | 5.6 ± 0.4 | 6.4 ± 0.2 | 5.2 ± 0.5 | 5.5 ± 0.5 | 4.4 ± 0.6 | + |

**Table 4  Female descriptive morphometric statistics.** Descriptive morphometric statistics (in mm) for females of nominal and CCS of *Amazophrynella*. KW = Kruskal Wallis test, (+) *p*-value < 0.05.

| Variable | *A. minuta* (*n* = 20) | *A. matses* (*n* = 13) | *A. javierbustamantei* (*n* = 28) | *A. moisesii* sp. nov (*n* = 15) | *A. amazonicola* (*n* = 15) | *A. siona* sp. nov. (*n* = 35) | *A. bokerma-nni* (*n* = 7) | *A. xinguensis* sp.nov. (*n* = 13) | *A. manaos* (*n* = 27) | *A. teko* sp. nov. (*n* = 17) | *A. vote* (*n* = 14) | KW *p-value* |
|---|---|---|---|---|---|---|---|---|---|---|---|---|
| SVL | 17.4 ± 0.9 | 17.1 ± 0.7 | 19.7 ± 1.8 | 18.5 ± 1.6 | 18.1 ± 1.1 | 18.3 ± 0.9 | 23.4 ± 0.8 | 24.1 ± 1.2 | 20.8 ± 2.1 | 19.2 ± 1.1 | 16.3 ± 1.6 | + |
| HW | 5.1 ± 0.4 | 4.8 ± 0.4 | 5.0 ± 0.3 | 5.1 ± 0.3 | 5.1 ± 0.4 | 5.1 ± 0.3 | 6.4 ± 0.3 | 6.3 ± 0.3 | 6.0 ± 0.6 | 5.4 ± 0.3 | 4.8 ± 0.4 | + |
| HL | 6.0 ± 0.4 | 5.6 ± 0.3 | 6.2 ± 0.3 | 6.4 ± 0.4 | 6.1 ± 0.4 | 6.2 ± 0.3 | 7.9 ± 0.3 | 7.9 ± 0.3 | 7.2 ± 0.3 | 6.5 ± 0.3 | 5.4 ± 0.4 | + |
| SL | 2.7 ± 0.2 | 2.7 ± 0.3 | 2.8 ± 0.2 | 2.9 ± 0.3 | 1.5 ± 0.2 | 2.9 ± 0.3 | 3.6 ± 0.1 | 3.75 ± 0.2 | 3.3 ± 0.3 | 2.9 ± 0.2 | 2.6 ± 0.3 | + |
| ED | 1.7 ± 0.3 | 1.4 ± 0.2 | 1.5 ± 0.3 | 1.9 ± 0.2 | 1.4 ± 0.1 | 1.7 ± 0.2 | 2.2 ± 0.2 | 2.1 ± 0.1 | 1.8 ± 0.2 | 1.8 ± 0.1 | 1.7 ± 0.2 | + |
| IND | 1.4 ± 0.1 | 1.2 ± 0.2 | 1.2 ± 0.1 | 1.4 ± 0.1 | 1.2 ± 0.1 | 1.4 ± 0.1 | 1.6 ± 0.1 | 1.6 ± 0.1 | 2.0 ± 0.1 | 1.5 ± 0.1 | 1.3 ± 0.1 | + |
| UAL | 5.2 ± 0.2 | 5.2 ± 0.2 | 6.1 ± 0.6 | 6.0 ± 0.5 | 5.5 ± 0.6 | 5.6 ± 0.4 | 7.9 ± 0.3 | 8.0 ± 0.4 | 5.5 ± 0.3 | 6.1 ± 0.5 | 4.9 ± 0.7 | + |
| HAL | 3.6 ± 0.3 | 3.7 ± 0.3 | 4.6 ± 0.4 | 4.6 ± 0.5 | 3.9 ± 0.4 | 3.9 ± 0.3 | 4.9 ± 0.2 | 5.0 ± 0.4 | 4.4 ± 0.3 | 4.1 ± 0.3 | 3.4 ± 0.5 | + |
| THL | 8.5 ± 0.9 | 8.3 ± 0.4 | 9.6 ± 0.8 | 9.8 ± 0.4 | 9.5 ± 0.8 | 9.4 ± 0.6 | 11.8 ± 0.7 | 11.8 ± 0.8 | 10.2 ± 0.6 | 9.5 ± 0.5 | 7.7 ± 0.8 | + |
| TAL | 8.4 ± 0.7 | 8.3 ± 0.4 | 9.8 ± 0.8 | 9.6 ± 0.5 | 9.1 ± 0.7 | 9.2 ± 0.6 | 11.0 ± 0.4 | 11.2 ± 0.6 | 10.2 ± 0.6 | 9.4 ± 0.6 | 7.2 ± 1.0 | + |
| TL | 5.4 ± 0.4 | 5.3 ± 0.4 | 5.9 ± 0.5 | 5.7 ± 0.3 | 5.4 ± 0. | 5.7 ± 0.5 | 6.9 ± 0.4 | 7.1 ± 0.4 | 7.1 ± 0.9 | 5.7 ± 0.4 | 4.6 ± 0.6 | + |
| FL | 6.4 ± 0.7 | 6.2 ± 0.4 | 7.2 ± 0.7 | 7.3 ± 0.7 | 6.5 ± 0.6 | 7.0 ± 0.6 | 8.6 ± 0.5 | 8.9 ± 0.5 | 8.1 ± 0.6 | 7.2 ± 0.62 | 5.6 ± 0.9 | + |

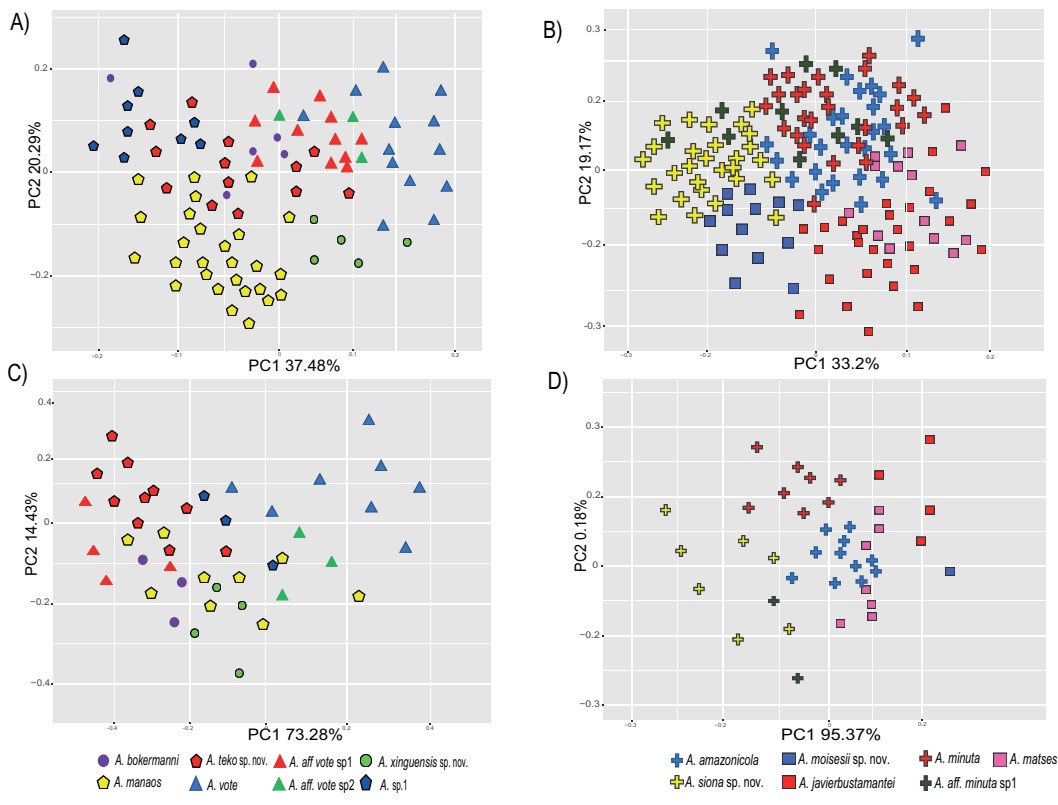

**Figure 3  Principal components analyses of morphometric and environmental variables.** Morphometric PCA: (A) Eastern clade, (B) Western clade. Environmental PCA: (C) Eastern clade, (D) Western clade. Symbols and colors represent the clades recovered by the phylogenetic analyses (Fig. 1). UCS and UL were not included.

by the PCA account for 52.37% of the variation found in the dataset. The first component (PC1) explained 33.2% of the total variation and the second component (PC2) explained 19.17% of the variation. Using the DFA a total of 68% of specimens were correctly assigned to phylogenetic groups. The number of individuals correctly assigned to each clade by DFA are presented in Table 5. The DFA showed that the variables that most contributed to the morphometric separation were eye diameter, hand length, head width and foot length. Head traits (head length, eye diameter and intranasal distance) and hand traits (hand length) were the variables that explained 78% of the classification by the first two discriminant axes (Appendix S8, Figs. S8A–S8B). Loadings and percentage of variance explained for discriminant axes (FI–II) of morphometric variables in western clade are provided in Appendix S8 and Table S8A.

## Environmental analyses
We obtained a total of 90 unique localities for final analysis, 43 localities of the eastern and 47 localities of the western clade, representing the occurrences of all species but *Amazophrynella* aff. *matses* sp, *A*.sp2 and *A* sp3 (see Fig. 1). The list of localities used

Table 5   Male classification in morphological space. Successful classification in morphological space (males) recovered phylogenetic mt DNA lineages (Eastern and Western clades). In parentheses, the percentage of successfully classification. The numbers in the cells represent the numbers of individuals assigned to each clade by discriminant analyses. UCS and UL were not included.

| Lineages (Eastern clade) | A. manaos (90%) | A. teko sp. nov. (68%) | A. vote (100%) | A. aff. vote sp1 (63%) | A. aff. vote sp2 (0%) | A. bokermanni (50%) | A. xinguensis sp. nov. (80%) |
|---|---|---|---|---|---|---|---|
| A. manaos | 27 | 0 | 0 | 0 | 0 | 0 | 0 |
| A. teko sp. nov. | 0 | 15 | 0 | 0 | 0 | 1 | 0 |
| A. vote | 0 | 0 | 13 | 0 | 0 | 0 | 0 |
| A. aff. vote sp1 | 1 | 1 | 0 | 7 | 2 | 0 | 0 |
| A. aff. vote sp2 | 0 | 0 | 0 | 3 | 0 | 0 | 0 |
| A. bokermanni | 1 | 1 | 0 | 0 | 0 | 3 | 1 |
| A. xinguensis sp. nov. | 1 | 0 | 0 | 0 | 0 | 0 | 4 |

| Lineages (Western clade) | A. matses (39%) | A. javierbustamantei (79%) | A. moisesii sp. nov. (31%) | A. amazonicola (85%) | A. siona sp. nov. (59%) | A. minuta (74%) | A. minuta sp1 (0%) |
|---|---|---|---|---|---|---|---|
| A. matses | 5 | 5 | 0 | 1 | 2 | 0 | 0 |
| A. javierbustamantei | 1 | 23 | 1 | 0 | 2 | 2 | 0 |
| A. moisesii sp. nov. | 0 | 0 | 4 | 0 | 7 | 0 | 2 |
| A. amazonicola | 0 | 1 | 0 | 22 | 2 | 1 | 0 |
| A. siona sp. nov. | 0 | 2 | 2 | 2 | 16 | 5 | 0 |
| A. minuta | 0 | 0 | 1 |  | 2 | 23 | 3 |
| A. minuta aff. sp1 | 0 | 0 | 2 | 0 | 0 | 7 | 0 |

for environmental analyses and discriminant function analyses are in Appendix S3 and Table S3A.

The PCA of the eastern and western clades revealed a grouping of specimens based on environmental traits and allowed us to distinguish all the mtDNA lineages in the multivariate space (Figs. 3C and 3D). Character loadings, eigenvalues and percentage of variance explained for PCA (PC 1–2) analyses for environmental variables for the eastern and western clades are provided in Appendix S7 and Tables S7C–S7D.

In the eastern clade specimens of each lineages can be successfully separated based on environmental traits using PCA (Fig. 3C). The first two principal components extracted by the PCA account for 87.71% of the variation found in the dataset. The first component (PC1) explained 73.28% of the total variation and the second component (PC2) explained 14.43% of the variation. A total of 65% of specimens were correctly classified to their lineage. The numbers of individuals correctly assigned to each clade by DFA are presented in Table 6. The environmental variables that most contributed to separating lineages were mean temperature of the coldest quarter (bio11), maximum temperature of the warmest month (bio5), mean diurnal temperature range (bio2) and isothermality (bio3) (Appendix S8, Figs. S8A–S8C). Loadings and percentage of variance explained per discriminant axes (F1–2) of environmental variables in the eastern clade are provided in Appendix S8 and Table S8B.
Table 6  **Male classification in environmental space.** Successful classification in environmental space recovered phylogenetic mt DNA lineages (Eastern and Western clades). In parentheses, the percentage of successful classifications. The numbers in the cells represent the numbers of individuals assigned to each clade by discriminant analyses. UCS and UL were not included.

| Lineages (Eastern clade) | A. manaos (77%) | A. teko sp. nov. (90%) | A. vote (80%) | A. aff. vote sp1 (40%) | A. aff. vote sp2 (33%) | A. bokermanni (50%) | A. xinguensis sp. nov. (66%) |
|---|---|---|---|---|---|---|---|
| A. manaos | 7 | 0 | 0 | 0 | 0 | 1 | 0 |
| A. teko sp. nov. | 0 | 10 | 0 | 0 | 0 | 0 | 0 |
| A. vote | 0 | 0 | 4 | 2 | 2 | 0 | 0 |
| A. aff. vote sp1 | 1 | 0 | 1 | 2 | 2 | 0 | 0 |
| A. aff. vote sp2 | 0 | 0 | 0 | 1 | 1 | 1 | 0 |
| A. bokermanni | 1 | 1 | 0 | 0 | 0 | 2 | 1 |
| A. xinguensis sp. nov. | 0 | 0 | 0 | 0 | 0 | 1 | 2 |

| Lineages (Western clade) | A. matses (87%) | A. javierbustamantei (100%) | A. moisesii sp. nov. (62%) | A. amazonicola (100%) | A. siona sp. nov. (80%) | A. minuta (70%) | A. minuta sp1 (0%) |
|---|---|---|---|---|---|---|---|
| A. matses | 7 | 0 | 0 | 0 | 0 | 0 | 0 |
| A. javierbustamantei | 0 | 6 | 2 | 0 | 0 | 0 | 0 |
| A. moisesii sp. nov. | 0 | 0 | 5 | 0 | 0 | 0 | 0 |
| A. amazonicola | 1 | 0 | 0 | 6 | 0 | 0 | 0 |
| A. siona sp. nov. | 0 | 0 | 0 | 0 | 8 | 2 | 1 |
| A. minuta | 0 | 0 | 1 | | 1 | 7 | 0 |
| A. aff. minuta sp1 | 0 | 0 | 0 | 0 | 1 | 1 | 0 |

In the western clade specimens of each lineages can be successfully separated based on environmental traits using PCA (Fig. 3D). The first two principal components extracted by the PCA account for 95.55% of the variation found in the dataset. The first component (PC1) explained 95.37% of the total variation and the second component (PC2) explained 0.18% of the variation. A total of 81% of specimens were correctly assigned to their candidate species. The numbers of individuals correctly assigned to each clade by DFA are presented in Table 6. The environmental variables that most contributed to group separation were annual mean temperature (bio1), mean diurnal temperature range (bio2), mean temperature of the warmest quarter (bio10) and mean temperature of the wettest quarter (bio8) (Appendix S8 and Figs. S8A–S8D). Loadings and percentage of variance explained for discriminant axes (F1–2) of environmental variables in the western clade are provided in Appendix S8 and Table S8B.

## Taxonomic decisions

Our data analysis of *Amazophrynella* suggest the existence of 18 linages of which seven are nominal species, three Deep Conspecific Lineages, one Unconfirmed Candidate Species, three Uncategorized Lineages and four Confirmed Candidate Species (Table 2). The four CCSs presented at least one diagnostic morphological character, monophyly with a strong phylogenetic support using the standard DNA barcode 16S fragment (*Vences et al., 2005*) and divergence from its sister taxon at environmental and morphometric data. Based on these results, herein we described *A. teko* sp. nov., *A. siona* sp. nov., *A. xinguensis* sp. nov., and *A. moisesii* sp. nov.

## Species accounts

*Amazophrynella teko* sp. nov.
urn:lsid:zoobank.org:act:590F41D2-7138-42F8-8509-448602C2D040
*Amazonella* sp. Guianas (*Fouquet et al., 2012a*: 829, French Guiana [in part])
*Amazophrynella* sp. Guianas (*Fouquet et al., 2012b*: 68, French Guiana [in part])
*Amazophrynella* sp. Guianas (*Rojas et al., 2015*: 85, French Guiana [in part])
*Amazophrynella* sp1. (*Fouquet et al., 2015*: 365, French Guiana [in part])
*Amazophrynella* sp. aff. *manaos* (*Rojas et al., 2016*: 49, French Guiana [in part])

*Holotype* (Fig. 4). MNHN 2015.136, adult male, collected at Alikéné (3°13′07″N, 52°23′47″W), 206 m a.s.l., district of Camopi, French Guiana by J.P. Vacher on March 21, 2015.

*Paratypes.* Twenty-six specimens (males = 13; females = 13). French Guiana: District of Saint Laurent du Maroni: Mitaraka layon (2°14′09″N, 54°26′57″W) 330 m a.s.l., MNHN 2015.137, MNHN 2015.138, MNHN 2015.139, MNHN 2015.140 (adult males), MNHN 2015.141, MNHN 2015.142, MNHN 2015.143 (adult females), A. Fouquet and M. Dewynter between 23 and 28 February 2015; Pic Coudreau du Sud (2°15′14″N, 54°21′04″W) 360 m a.s.l., MNHN 2015.152 (adult male), MNHN 2015.153 (adult female), M. Blanc on February 2015. Flat de la Waki (3°05′15″N, 53°24′12″W) 173 m a.s.l., INPA–H 36598 (adult female), J.P. Vacher on April 04, 2014. District of Camopi: Mitan (2°37′42″N, 52°33′15″W) 110 m a.s.l., INPA–H 36596, MNHN 2015.144, MNHN 2015.145, MNHN 2015.146, MNHN 2015.147, MNHN 2015.148 (adult males), MNHN 2015.149, MNHN 2015.150 (adult females), A. Fouquet and P. Nunes between 20 and 24 March 2015. Alikéné (3°13′07″N, 52°23′47″W) 206 m a.s.l. District of Saint Georges: Saint Georges (3°58′03″N, 51°52′20″W) 76 m a.s.l., MNHN 2015.151 (adult male), A. Fouquet and E. Courtois on February 2015; Mémora (3°18′47″N, 52°10′49″W) 77 m a.s.l., MNHN 2015.154 (adult male), MNHN 2015.155 (adult female), A. Fouquet and P. Nunes on March 18, 2015; Saut Maripa (3°48′22″N, 51°53′36″W) 51 m a.s.l., INPA–H 36597, INPA–H 36610, INPA–H 36599, INPA–H 36601, INPA–H 36600 (adult females), Antoine Fouquet and E. Courtois on February 2012.

*Diagnosis.* An *Amazophrynella* with (1) SVL12.9–15.8 mm in males, 17.9–21.5 mm in females; (2) snout acute in lateral view; upper jaw, in lateral view, protruding beyond lower jaw; (3) texture of dorsal skin granular; (4) cranial crest, vocal slits and nuptial pads absent; (5) dorsum covered by abundant rounded granules; (6) abundance of granules on tympanic area, on edges of upper arms and on dorsal surface of arms; (7) ventral skin highly granular; (8) fingers slender, basally webbed; (9) finger III relatively short (HAL/SVL 0.2–0.22 mm, $n = 30$); (10) finger I shorter than finger II; (11) palmar tubercle protruding and elliptical; (12) hind limbs relatively short (TAL/SVL 0.48–0.49, $n = 30$); (13) toes slender, basally webbed; in life: (14) venter cream; small blotches on venter.

*Comparison with other species (characteristics of compared species in parentheses).* *Amazophrynella teko* sp. nov. is morphologically most similar to *A. manaos* from which it can be distinguished by: large SVL of males 12.9–15.8 mm, $n = 13$ (vs. 12.3–15.0 mm,

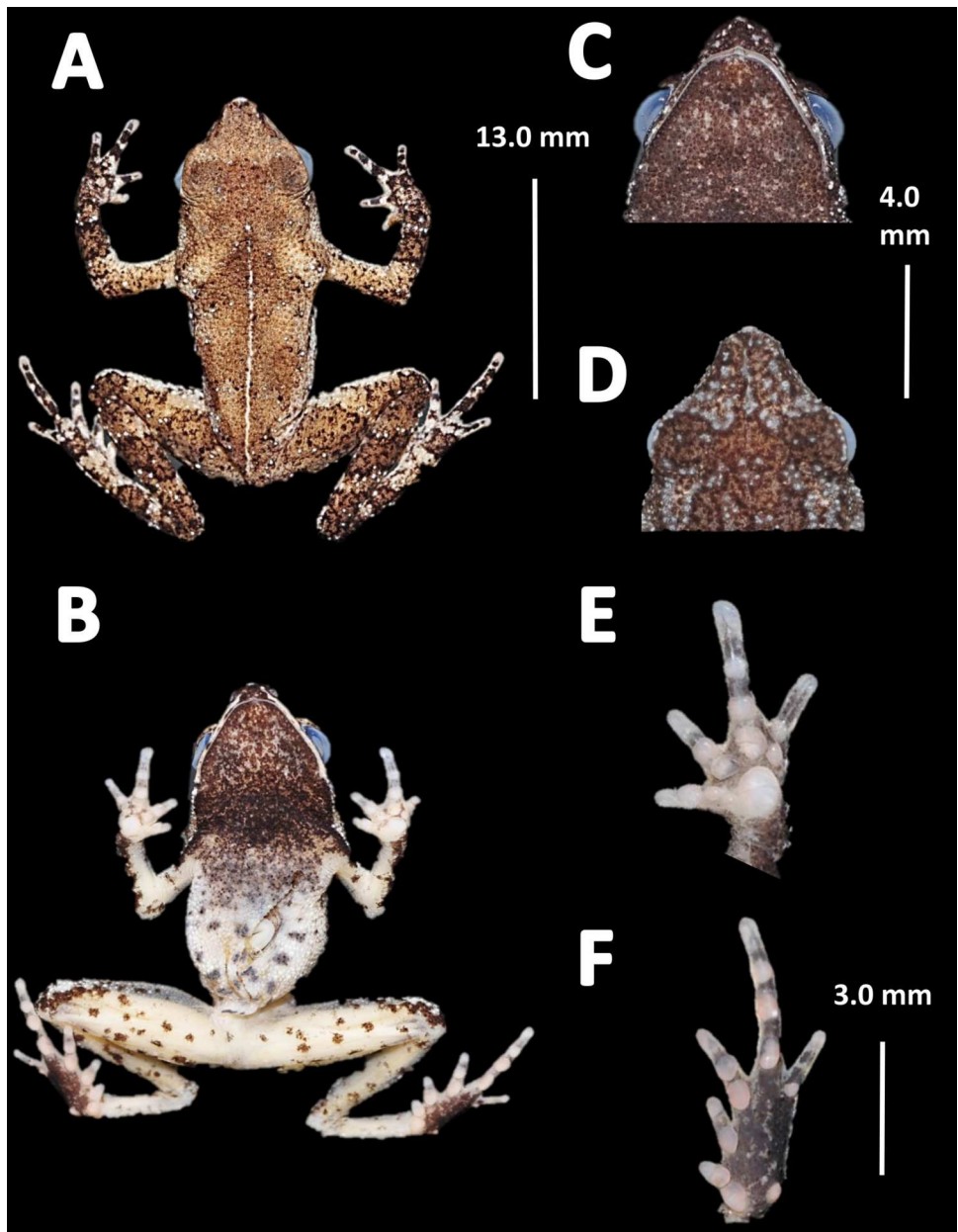

**Figure 4** **Holotype of *Amazophrynella teko*. sp. nov. (MNHN 2015.136).** (A) Dorsal view; (B) ventral view; (C) dorsal view of the head; (D) ventral view of the head; (E) left toe; (F) left hand. Photos by Rommel R. Rojas.

$n = 27$, Fig. 5, $t = 2.04$, $df = 16.78$, $p$-value = 0.02); snout acute in lateral view (truncate); larger THL of males, 53% of SVL, $n = 13$ (vs. smaller THL, 47.2% of SVL, $n = 27$); abundance of granules on tympanic area (absent); smaller hind limbs, TAL/SVL 0.48–0.49, $n = 30$ (vs. 0.50–0.51, $n = 56$). From *A. bokermanni* by the relative size of fingers: FI < FII (vs. FI > FII); thumb not large and robust (thumb large and robust, Figs. 6A vs. 6D). From *A. vote* by larger SVL of males 12.9–15.8 mm, $n = 13$ (vs. 10.0–14.2 mm, $n = 14$,

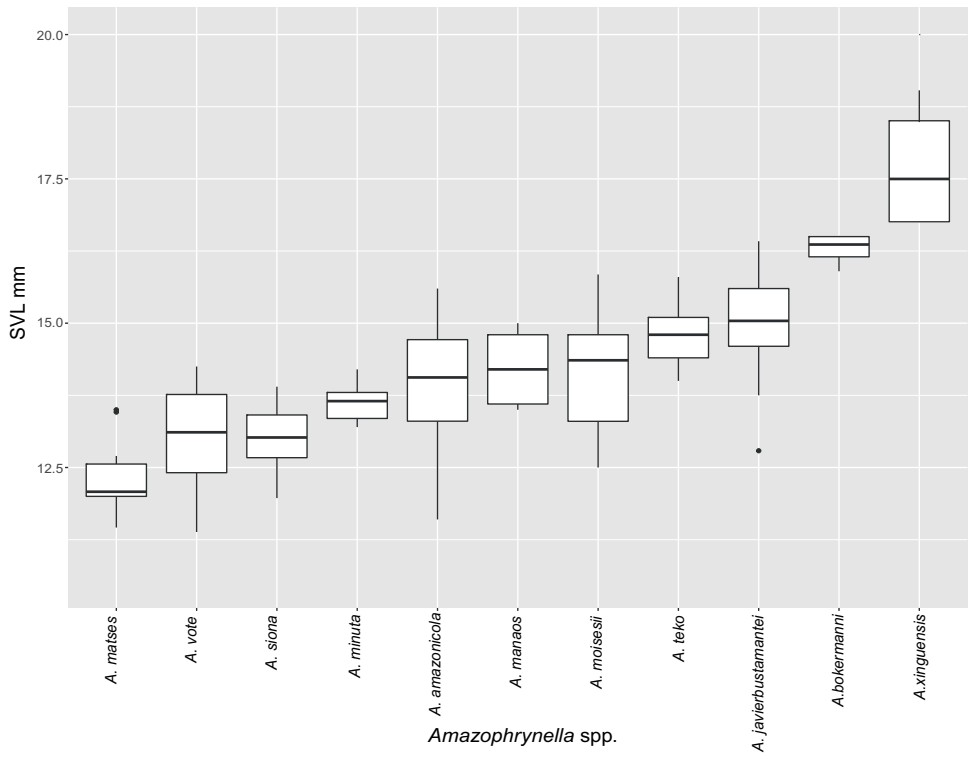

**Figure 5 Measurement comparison of SVL between males of nominal species of *Amazophrynella*.**

see Fig. 3, $t = 4.93$, $df = 25.91$, $p$-value $= 0.001$) and females 17.9–21.5 mm, $n = 17$ (vs. 13.5–19.1 mm, $n = 21$); texture of dorsal skin granular (tuberculate); longer UAL, 33% of SVL (vs. smaller UAL 29.8%); longer hind limbs, TAL/SVL 0.48–0.49, $n = 30$ (vs. 0.43–0.44, $n = 35$); venter coloration cream (red-brown, Figs. 7B vs. 7F. From *A. minuta* by snout acute in lateral view (pointed, Figs. 8A vs. 8B); larger snout of males–50% of HL, $n = 14$ (vs. SL 46% of HL, $n = 13$); palmar tubercle elliptical (rounded, Figs. 6A vs. 6G); venter cream (yellow-orange, Figs. 7A vs. 7B). From *A. amazonicola* by dorsal skin texture granular (finely granular); absence of small triangular protrusion on the tip of the snout (present, Figs. 8A vs. 8H); palmar tubercle elliptical (rounded); venter coloration cream (venter yellow–orange). From *A. matses* by smaller SVL of males 12.9–15.8 mm, $n = 13$ (vs. 11.4–13.5 mm, $n = 13$, Table 3 and Fig. 3, $t = 7.89$, $df = 21.34$, $p$-value $= 0.001$) and females 17.9–21.5 mm, $n = 17$ (vs. 15.6–19.0 mm, $n = 18$); snout profile acute in lateral view (truncate); texture of dorsal skin granular (spiculate); venter cream (venter pale yellow). Compared to *A. javierbustamantei* by shorter hand, HAL/SVL 0.2–0.22, $n = 30$ (vs. 0.23–0.24, $n = 60$); texture of dorsal skin granular (tuberculate); venter cream (pale orange yellowish); tiny blotches on venter (tiny rounded points, Figs. 7B vs. 7J). Compared to *A. siona* sp. nov. by large size SVL of adult males 12.9–15.8 mm, $n = 14$ (vs. 11.5–14.7 mm, $n = 27$, Fig. 5, $t = 6.15$, $df = 18.1$, $p$-value $= 0.001$) and adult females 17.9–21.5 mm, $n = 17$, (vs. 16.1–20.0 mm, $n = 35$) and; smaller hind limbs, TAL/SVL 0.48–0.49, $n = 30$ (vs. 0.5–0,52, $n = 62$); palmar tubercle elliptical (rounded), venter cream (venter bright

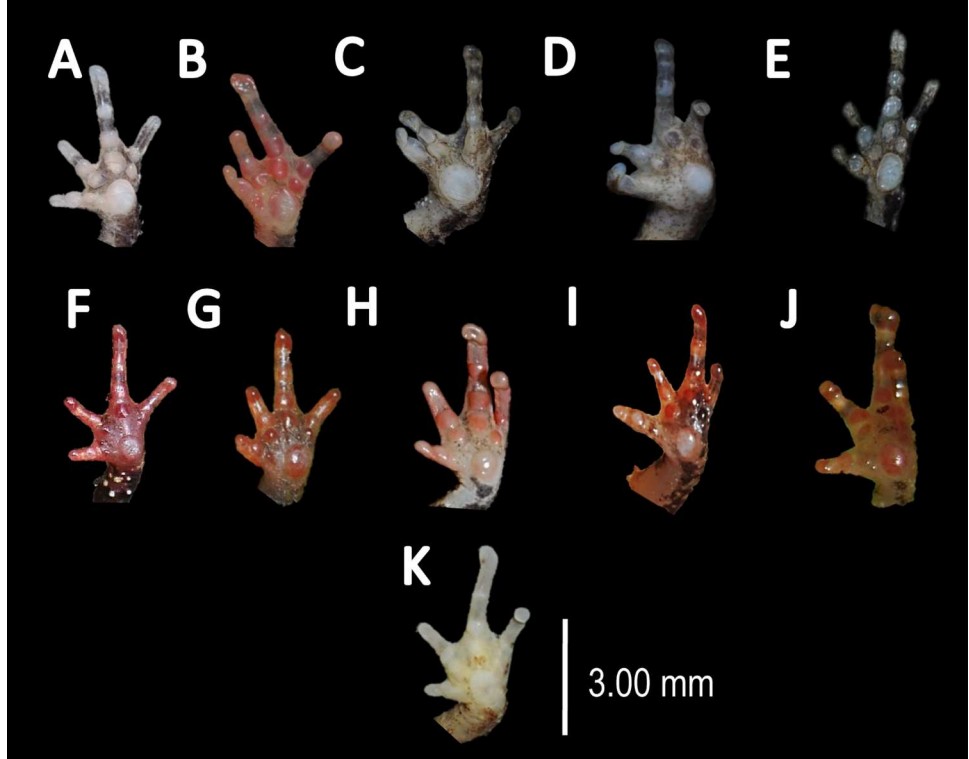

**Figure 6   Comparison of palmar tubercles of nominal species of *Amazophrynella*.** (A) *A. teko* sp. nov. (B) *A. siona* sp. nov. (C) *A. xinguensis* sp. nov. (D) *A. bokermanni.* (E) *A. vote.* (F) *A. amazonicola.* (G) *A. minuta.* (H) *A. matses.* (I) *A. manaos.* (J) *A. javierbustamantei.* (K) *A. moisesii* sp. nov. Elliptical (A, I, J); rounded (B, E, D, H, F, G); ovoid (C). See Table 2. Photos by Rommel R. Rojas.

red). From *A. xinguensis* sp. nov. by FI <FII (vs. FI ≥ FII, Figs. 6A vs. 6C; palmar tubercle rounded (ovoid). From *A. moisesii* sp. nov. by venter cream (venter pale yellow); shorter hand, HAL/SVL 0.2–0.22, $n = 30$ (vs. 0.23–0.25, $n = 28$).

*Description of the holotype.* Body slender, elongate. Head triangular in lateral view and pointed in dorsal view. Head longer than wide. HL 34.4% of SVL. HW 27.8% of SVL. Snout acute in lateral view and triangular in ventral view. SL 50% of HL. Nostrils slightly protuberant, closer to snout than to eyes. *Canthus rostralis* straight in dorsal view. Internarial distance smaller than eye diameter. IND 33.3% of HW. Upper eyelid covered with smaller pointed tubercles. Eyes wide, prominent, ED 30.7% of HW. Tympanum not visible through the skin. Skin around tympanum covered by granules. Vocal sac not visible. Texture of dorsal skin granular. Texture of dorsolateral skin granular. Forelimbs slender. Edges of forelimbs with scattered granules, in dorsal and ventral view. Upper arms robust. UAL 33.1% of SVL. Abundance of granules on upper arm. HAL about 22.5% of UAL. Fingers basally webbed. Fingers slender, tips unexpanded. Relative length of fingers: I<II<IV<III. Supernumerary tubercles and accessory palmar tubercles rounded. Palmar tubercle small and rounded. Subarticular tubercles rounded. Texture of gular region granular. Texture of ventral skin highly granular. Small granules in the venter.

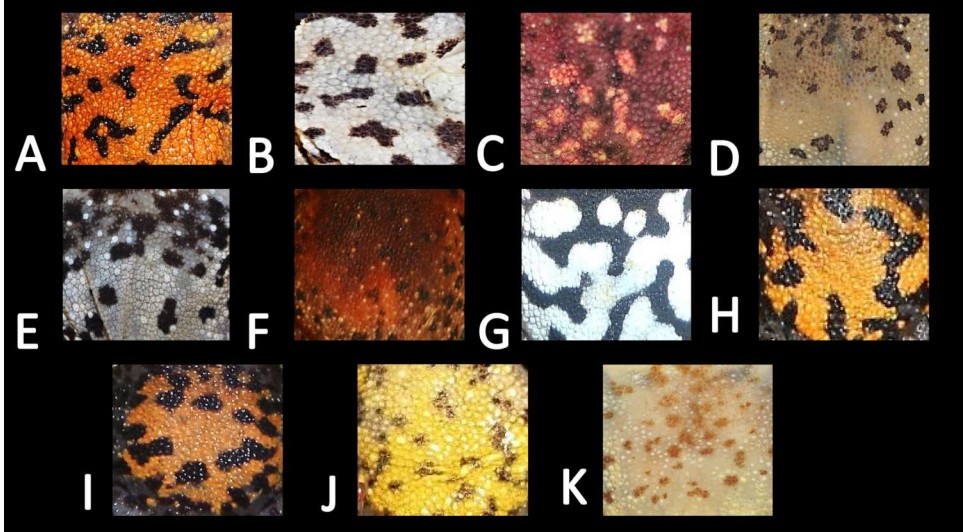

**Figure 7** **Ventral skin coloration of *Amazophrynella* spp.** Ventral skin coloration of nominal species of *Amazophrynella*. (A) *A. minuta*. (B) *A. teko* sp. nov. (C) *A. siona* sp. nov. (D) *A. xinguensis* sp. nov. (E) *A. bokermanni*. (F) *A. vote*. (G) *A. manaos*. (H) *A. amazonicola*. (I) *A. matses*. (J) *A. javierbustamantei*, (K) *A. moisesii* sp. nov. Large blotches (A, G); medium size blotches (H); small blotches (B, I, C); small dots (F, E, J); medium size dots (D); tiny points (K). See Table 2. Photos by Rommel R. Rojas.

Hindlimbs slender. Edges of the thigh to tarsus covered by conical tubercles. THL 52.3% of SVL. TAL 45.6% of SVL. Tarsus slender. TL 29.8% of SVL. FL 70.8%. Relative length of toes: I<II<III<V<IV. Inner metatarsal tubercle oval. Outer metatarsal tubercles small and rounded. Subarticular tubercles rounded. Toes slender and elongate. Tip of toes not expanded, basally webbed. Cloacal opening slightly above midlevel of thighs.

*Measurement of the holotype* (*in mm*). SVL: 15.1; HW: 4.2; HL: 5.2; SL: 2.6; ED: 1.6; IND: 1.4; UAL: 5.0; HAL: 3.4; THL: 7.9; TAL: 6.9; TL: 4.5; FL: 5.6.

*Variation* (Fig. 9). There is little variation among the examined specimens. Sexual dimorphism was observed in SVL, with 12.9–15.8 mm (14.7 $\pm$ 0.8 mm, $n = 13$) in males and 17.9–21.5 mm (19.2 $\pm$1.8 mm, $n = 17$) in females. Specimens (MNHN 2015.137, MNHN 2015.138, MNHN 2015.139, MNHN 2015.140) present lesser abundance of granules on arm insertion. In some individuals (MNHN 2015.143) the ventral and the dorsolateral region present one to three large tubercles. Subarticular tubercles more protruding and swollen in females. Blotches on belly display different sizes (larger vs. small, see Fig. 10). In life, venter coloration between cream to off-white . Palm and sole between light red and orange. In preserved specimens, the palmar tubercle is more flattened than in life.

*Coloration of the holotype (in life)*. Head black brown, in dorsal view. Dorsum brown. Flanks brown. Scattered tubercles on flanks white. Dorsal surfaces of upper arm, arm and hand black. Dorsal surfaces of thighs, tibia, tarsus and foot black. Ventral surfaces of upper arm, arm and palm cream. Ventral surfaces of thighs cream, mottled with black blotches. In dorsal view, tarsus and tibia creamy, sole light red. Gular region brown. Belly cream

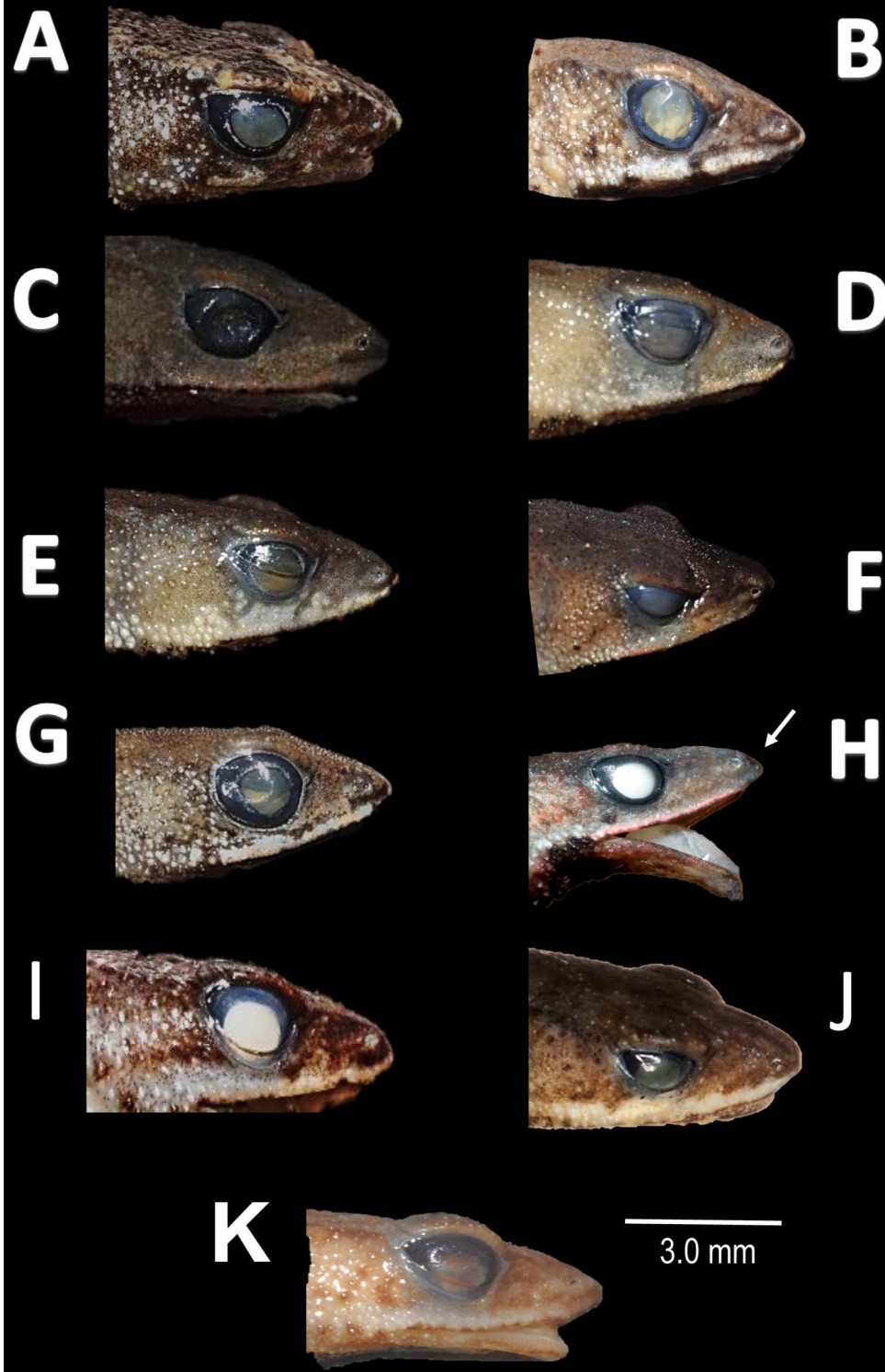

**Figure 8** **Comparison of head profile of nominal species of *Amazophrynella* in lateral view.** (A) *A. minuta*. (B) *A. teko* sp. nov. (C) *A. siona* sp. nov. (D) *A. xinguensis* sp. nov. (E) *A. bokermanni*. (F) *A. vote*. (G) *A. manaos*. (H) *A. amazonicola*. (I) *A matses*. (J) *A. javierbustamantei*. (K) *A. moisesii* sp. nov. Arrow indicates a small protuberance in the tip of the snout of *A. amazonicola*. Pointed (A, H, D, E); acute (B, C, I); truncate (G); rounded (F); acuminate (K, J). See Table 2. Photos by Rommel R. Rojas.

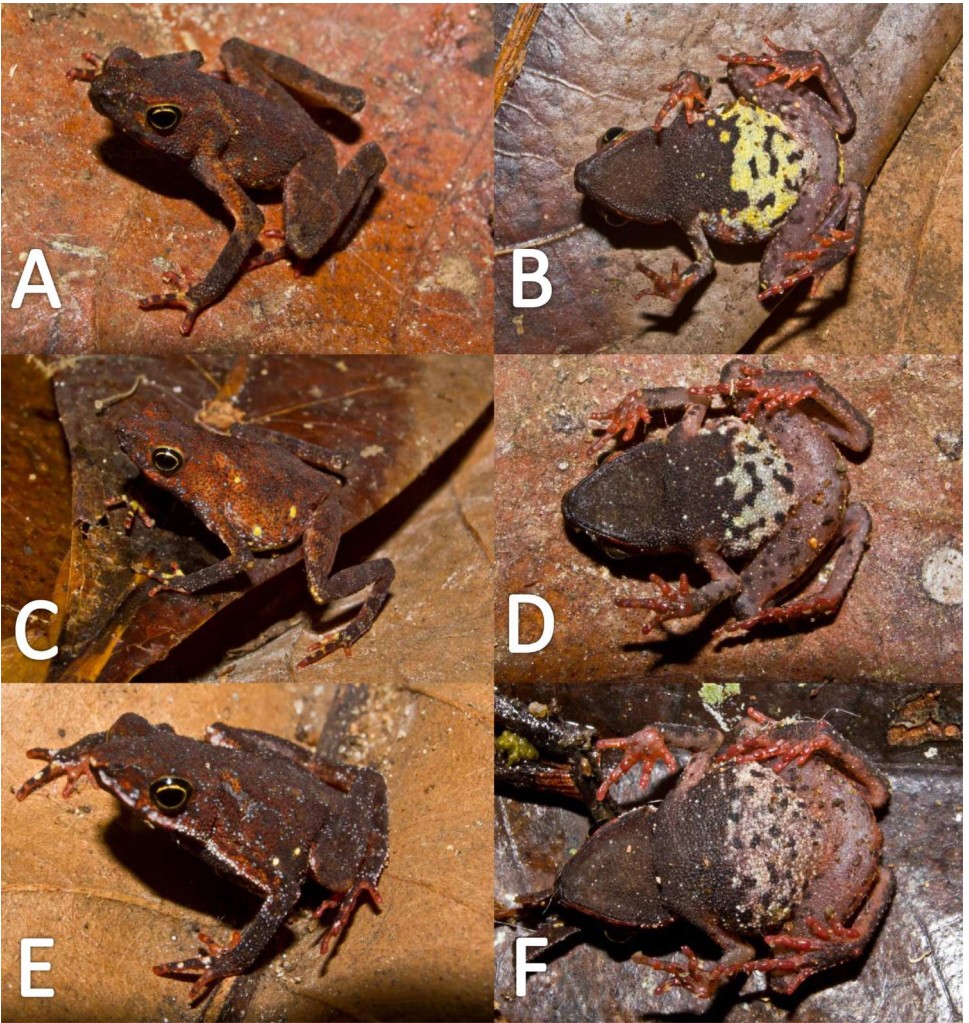

Figure 9 **Morphological variation in live *Amazophrynella teko* sp. nov. (unvouchered specimens).**
Adult males (A–D); adult females (E–F). Photos by Antoine Fouquet.

with black tiny blotches. Posterior region of the thigh and cloaca with black blotches. Longitudinal white stripe on upper jaw extending from nostril to tympanum. Iris golden and pupil black.

*Color in preservative* (~*70% ethanol*, Fig. 10). Almost the same as color in life. We noted the progressive loss of dorsal coloration which eventually becomes black. The chest lost its coloration and became less intense. The dark blotches on venter became less evident. The coloration of the fingers and toes became pale red.

*Bioacoustics* (Fig. 11). *Lescure & Marty (2000)* described the advertisement call of *Amazophrynella teko* sp. nov. as the call of *Dendrophryniscus minutus*. We recorded two individuals at Mitaraka (2°14′09″N, 54°26′57″W) and Alikéné (3°13′07″N, 52°23′47″W),

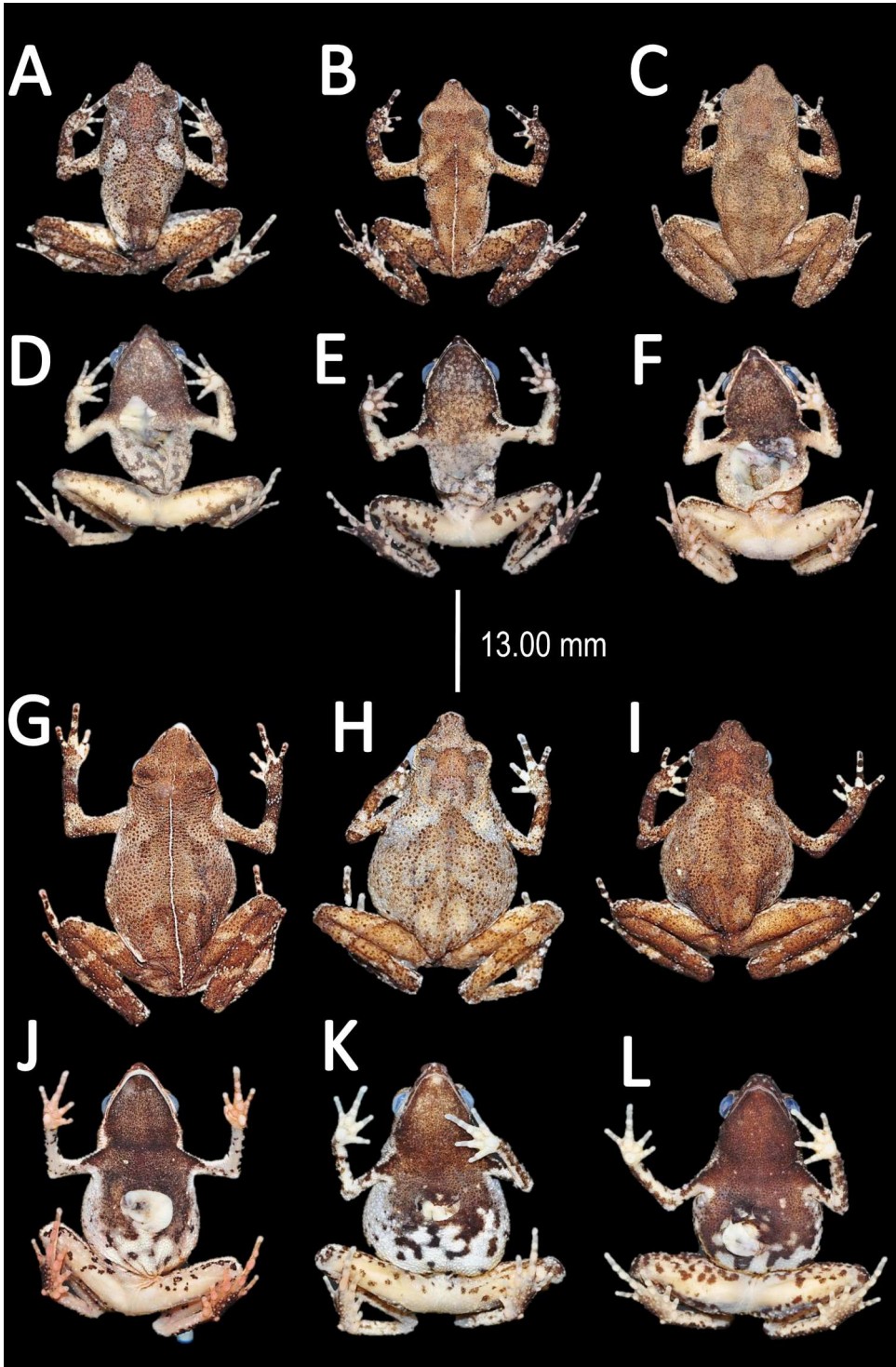

**Figure 10** **Morphological variation of preserved specimens of *Amazophrynella teko* sp. nov.** Adult males: MHNN 2015.138 (A–B); MHNN 2015.152 (C–D); MHNN 2015.139 (E–F). (G–L) Adult females: MHNN 2015.141 (G–H); MHNN 2015.143 (I–J); MHNN 2015.150 (K–L). Photos by Rommel R. Rojas.

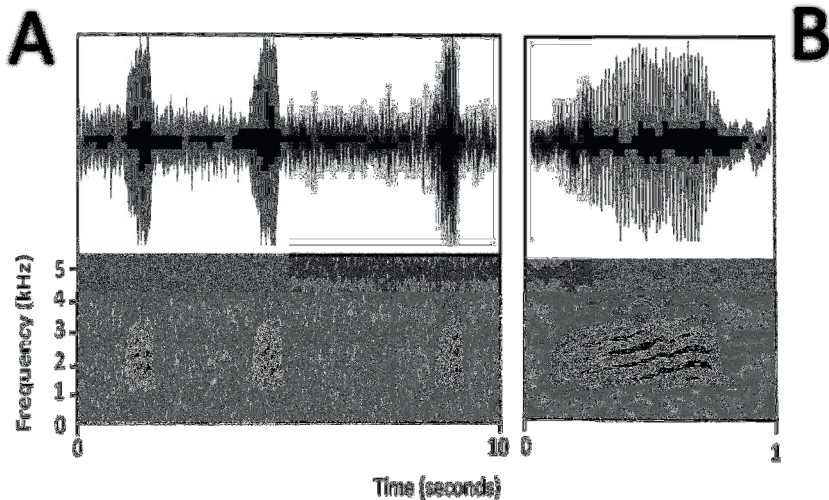

**Figure 11 Oscillogram and spectrogram of the advertisement call of *Amazophrynella teko* sp. nov.** (A) three notes; (B) one note.

French Guiana. All call parameters described by *Lescure & Marty (2000)* show an overlap with our recorded calls. Call trill emitted at regular intervals. Note duration 0.15–0.19 s (0.16 ± 0.01 s, $n = 29$). Fundamental frequency between 2,733.3–3,555.3 Hz (3,115.3 ± 263.7 Hz, $n = 29$). Dominant frequency between 3,993.3–4,980.8 Hz (4638.4 ± 288.27 Hz, $n = 29$). Number of pulses between 10–30 per call (25.5 ± 10.4 pulses/call, $n = 29$). Time to peak amplitude between 0.06–0.13 s (0.08 ± 0.02 s, $n = 29$). The call has a downward modulation, reaching its maximum frequency near its beginning.

*Distribution and natural history* (Fig. 1B). *Amazophrynella teko* sp. nov. have been recorded from the district of Saint Laurent du Marioni, Saint Georges and Camopi, French Guiana, the state of Amapá, Brazil and in the southern region of Suriname (A Fouquet, pers. obs., 2017). It occurs at elevations ranging from 70 m a.s.l. to 350 m a.s.l. The species is diurnal and crepuscular but is also active at night during peak breeding period, which normally occurs at the beginning of the rainy season (January–February). This species shows a conspicuous sexual dimorphism, with males being much smaller than females. The conservation status of this species remains unknown. The habitat destruction and pollution must affect their populations; however, due to its abundance we believe that this species probably needs not be classified above Least Concern category.

*Etymology.* The specific epithet is a noun in apposition and refers to the name of the Teko Amerindians who occupy the southern half of French Guiana; the area occupied by the Teko tribe also encompasses the type locality.

*Amazophrynella siona* sp. nov.
urn:lsid:zoobank.org:act:66224D58-8DE0-4D5B-950D-1206FFA4AC11

*Atelopus minutus*: (*Duellman & Lynch, 1969*: 238, Sarayacu [Ecuador])

*Dendrophryniscus minutus* (*Duellman, 1978*: 120, Santa Cecilia [Ecuador])

*Dendrophryniscus minutus* (Duellman & Mendelson III 1995: 336, vicinities of San Jacilllo and Teniente Lopez [Peru])

*Amazonela* cf. *minutus* "western Amazonia" (*Fouquet et al., 2012a*: 829, "western Amazonia", Ecuador [in part])

*Amazophrynella* cf. *minutus* "western Amazonia" (*Fouquet et al., 2012a*: 68, "western Amazonia", Ecuador [in part])

*Amazophrynella* aff. *minuta* "western Amazonia" (*Rojas et al., 2015*: 84, "western Amazonia", Ecuador [in part])

*Amazophrynella* aff. *minuta* (*Rojas et al., 2016*: 49, "western Amazonia", Ecuador [in part])

*Holotype* (Fig. 12). QCAZ 27790, adult male, collected at Yasuni National Park, (0°40′01″S, 76°26′33″W), 200 m a.s.l., Bloque 31, Apaika, Province of Orellana, Ecuador, by F. Nogales on October 7 2000.

*Paratypes.* Sixty-six specimens (males = 17, females = 49), Ecuador: Provincia Sucumbíos: Reserva de Producción Faunística Cuyabeno (0°00′58″S, 76°09′59″W), 203 m a.s.l., QCAZ 52433–34, S. R. Ron; Reserva de Producción Faunística Cuyabeno (0°00′58″S, 76°09′59″W), 203 m a.s.l, QCAZ 37758–59, QCAZ 37761, L. A. Coloma; Reserva de Producción Faunística Cuyabeno (0°00′58″S, 76°09′59″W), 203 m a.s.l., QCAZ 6071, QCAZ 6091, QCAZ 6095, QCAZ 6097, QCAZ 6105 (adult females), QCAZ 6111 (adult males), QCAZ 6113, QCAZ 6118, QCAZ 6127, QCAZ 6128, J. P. Caldwell; Santa Cecilia (0°04′50″S, 76°59′24″W), 330 m a.s.l., QCAZ 4469, QCAZ 4472, M. Crump; Tarapoa (0°07′10″S, 76°20′23″W), 330 m a.s.l., QCAZ 36331, QCAZ 36336, QCAZ 36338, QCAZ 36357, E. Ponce. Provincia Pastaza: Community of Kurintza (2°03′50″S, 76°47′53″W), 350 m a.s.l., QCAZ 56342 (adult female), QCAZ 56354, QCAZ 56361 (adult males), D. Velalcázar; A. Villano community, AGIP oil company (1°30′28″S, 77°30′41″W), 307 m a.s.l., QCAZ 38599, QCAZ 38679, QCAZ 38722, Galo Díaz; Around Villano community, AGIP oil company (1°30′28″S, 77°30′41″W), 307 m a.s.l. QCAZ 38642, Y. Mera; Community of Kurintza (2°03′50″S, 76°47′53″W), 350 m a.s.l., QCAZ 38809 (adult females), F. Varela; Community of Kurintza (2°03′50″S, 76°47′53″W), 350 m a.s.l., QCAZ 54213, Yerka Sagredo; Bataburo Lodge (1°12′30″S, 76°42′59″W), 260 m a.s.l., QCAZ 39408 (adult female), S. D. Padilla; Lorocachi (1° 37′17″S, 75°59′21″W), 229 m a.s.l., QCAZ 8902 (adult female), M. C. Terán; Lorocachi (1°37′17″S, 75°59′21″W), 229 m a.s.l., QCAZ 56165 (adult male), S. R. Ron; Bloque 31 in Yasuni National Park, (0°56′20″S, 75°50′20″W), 230 m a.s.l, QCAZ 11973, QCAZ 11979, QCAZ 11981 (adult males), G. Fletcher; Canelos (0°29′53″W, 76°22′26″S), 265 m a.s.l., QCAZ 52819, QCAZ 52823, D. Pareja; Canelos (0°29′53″W, 76°22′26″S), 265 m a.s.l., QCAZ 17391, L. A. Coloma. Provincia Orellana: Tambococha (0°58′42″S, 75°26′13″W), 194 m a.s.l., QCAZ 55345 (adult female), Fernando

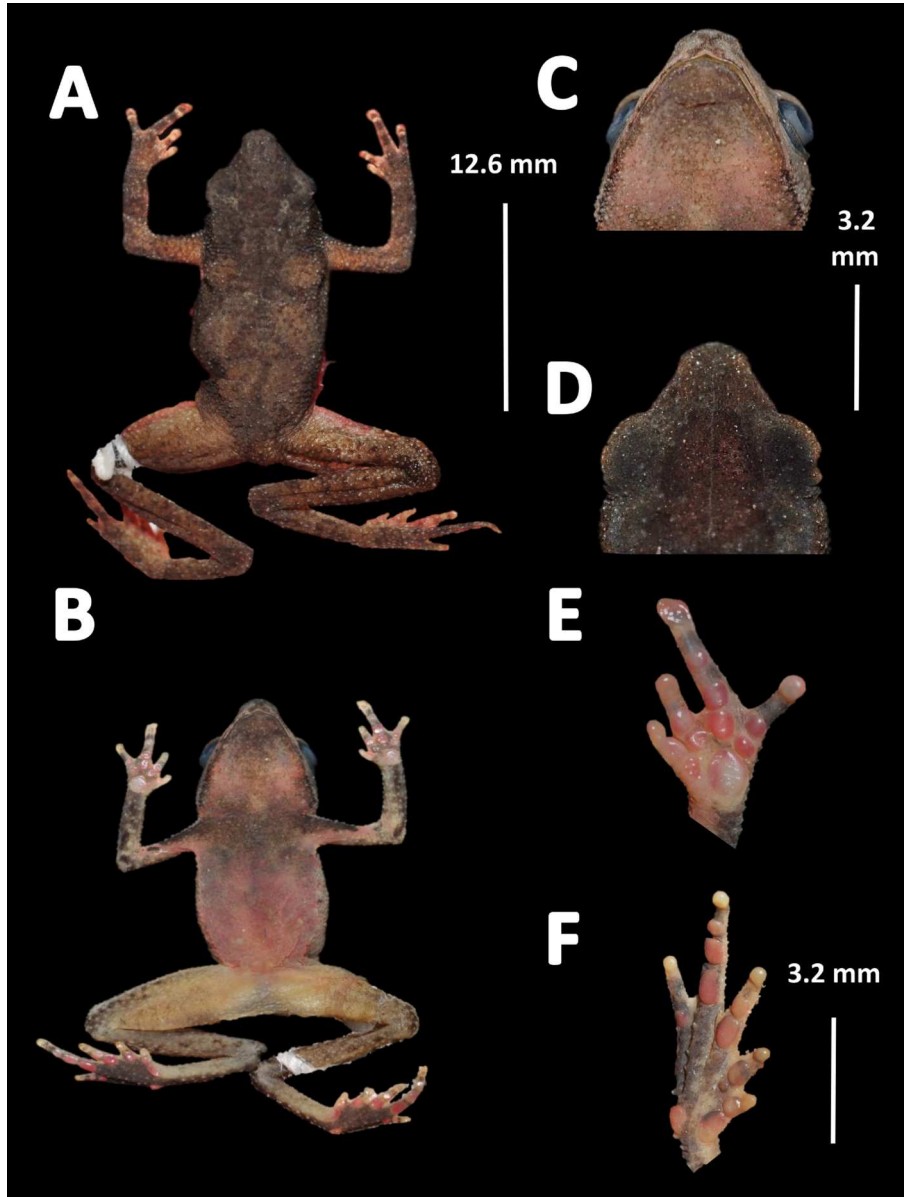

**Figure 12 Holotype of *Amazophrynella siona*. sp. nov. (QCAZ 27790).** (A) Dorsal view; (B) ventral view; (C) ventral view of head; (D) dorsal view of head; (E) right hand; (F) right foot. Photos by Rommel R. Rojas.

Ayala-Varela; Yasuni National Park, scientific station of the Pontificia Universidad Católica del Ecuador-PUCE, (0°56′31″S, 75°54′18″W), 203 m a.s.l., QCAZ 51068, E. Contreras; Yasuni National Park, scientific station of the Pontificia Universidad Católica del Ecuador-PUCE, (0°56′31″S, 75°54′18″W), 203 m a.s.l., QCAZ 21425, QCAZ 21431 (adult females), J. Santos; Garzacocha (0°45′28″S, 76°00′44″W), 230 m a.s.l., QCAZ 20504 (adult female), M. Díaz; Yuriti (0°33′26″S, 76°48′55″W), 220 m a.s.l., QCAZ 10526, (adult female), M. Read; Kapawi Lodge (2°32′19″S, 76°51′30″W), 257 m a.s.l., QCAZ 8725, S. R. Ron; Kapawi

Lodge (2°32′19″S, 76°51′30″W), 257 m a.s.l., QCAZ 25504 (adult males), QCAZ 25533 (adult female), K. Elmer; Fatima, 10 km from Puyo (1°24′47″S, 77°59′56″W), 1,000 m a.s.l., QCAZ 7135 (adult female), M. Tapia; Provincia Morona Santiago: Pankints (2°54′07″S, 77°53′39″W), 320 m a.s.l., QCAZ 46430 (adult female), J. B. Molina. Peru: Department Loreto: Teniente Lopez (2°35′30.90″S, 76°07′2.84″W), 255 m a.s.l., MUBI 7611, MUBI 7685, MUBI 7686, MUBI 7698, MUBI 7699, MUBI 7700 (adult females), J. C. Chaparro on October 12, 2008; Jibarito (2°47′55.90″S, 76°0′21.51″W), 236 m a.s.l., MUBI 7786, MUBI 7809, MUBI 7814 (adult female), J. Delgado on November 5, 2008; Shiviyacu (2°29′30.92″S, 76°5′18.31″W), 226 m a.s.l., MUBI 14730 (adult female), M. Medina on June 17, 2008; Jibarito (2°43′51.4″S, 76°01′7.48″W), near Corrientes River, 220 m a.s.l., MUBI 6292 (adult female), G. Chavez on March 20, 2008.

*Referred specimens.* USNM 520,898, 520900b–01 (adult males), USNM 520896–97, 520,899, 520901, 520906 (adult females), collected at Lagarto Cocha River (0°31′23″S, 75°15′25″W), Province of Loreto, Peru by S. W. Gotte on March 1994.

*Diagnosis.* An *Amazophrynella* with (1) SVL 11.5–14.7 mm in males, 16.1–20.0 mm in females; (2) snout acute in lateral view; upper jaw, in lateral view, protruding beyond lower jaw; (3) texture of dorsal skin finely granular; (4) cranial crests, vocal slits and nuptial pads absent; (5) small granules from the outer edge of the mouth to upper arm; (6) ventral skin granular; (7) tiny granules on ventral surfaces; (8) fingers slender, basally webbed; (9) finger III relative short (HAL/SVL 0.20–0.21, $n = 62$); (10) finger I shorter than finger II; (11) palmar tubercle rounded; (12) hind limbs relatively large (TAL/SVL 0.5–0.52, $n = 62$); (13) toes lacking lateral fingers; in life: (14) venter reddish brown; yellow blotches on venter.

*Comparison with other species (characteristics of compared species in parentheses).* *Amazophrynella siona* sp. nov. is most similar to *A. amazonicola* from which it can be distinguished by (characteristics of compared species in parentheses): the snout acute in lateral view (pointed, Figs. 8C vs. 8H), absence of protuberance on the tip of the snout (present); fingers basally webbed (webbing between FI and FII); yellow blotches on venter (dark blotches, Figs. 7C vs. 7H). From *A. matses* by the texture of dorsal skin granular (spiculate); larger HL, 5.6–7.2 mm in adult males, $n = 27$ (vs. 4.4–6.2 mm, $n = 26$, $t = 7.21$, $df = 20.1$, $p$-value $= 0.001$); snout acute in lateral (truncate); palmar tubercle rounded (elliptical, Figs. 6B vs. 6F; yellow blotches on venter (black blotches). From *A. minuta* by texture of dorsal skin finely granular (highly granular); small granules from the outer edge of the mouth to upper arm (small warts); tiny granules cover the venter surfaces (absent); shorter HAL, HAL/SVL 0.20–0.21, $n = 62$ (vs. 0.2–0.3, $n = 20$). Compared to *A. javierbustamantei* by shorter hand, HAL/SVL 0.20–0.21, $n = 62$ (vs. 0.23–0.24 , $n = 60$); texture of dorsal skin finely granular (finely tuberculate); snout acute in lateral view (subacuminate). From *A. bokermanni* by the relative size of fingers with FI<FII (FI>FII); thumb not large and robust (large and robust, Figs. 6B vs. 6D). From *A. vote* by snout acute in profile (rounded); dorsal skin finely granular (tuberculate); dorsal coloration light brown (brown); venter bright red (red-brown, Figs. 7C vs. 7F; yellow blotches on venter (white tiny spots)). From *A. manaos* by present rounded palmar tubercle (elliptical); snout acute in profile (truncate); venter bright red (white, Figs. 7C vs. 7G); yellow blotches on venter

(black patches). Compared to *A. teko* sp. nov. by small SVL of adult males 11.5–14.7 mm, $n = 27$ (12.9–15.8 mm, $n = 14$, $= 6.15$, $df = 18.1$, $p$ value $= 0.001$, Fig. 5) and adult females 16.1–20.0 mm, $n = 35$ (vs. 17.9–21.5 mm, $n = 17$); tiny granules cover venter (absent); longer hind limbs, TAL/SVL 0.5–0.52, $n = 62$ (vs. 0.48–0.49, $n = 30$); palmar tubercle round (elliptical); venter bright red (cream). From *A. xinguensis* sp. nov. by FI<FII (vs. FI $\geq$ FII, Fig. 6); palmar tubercle rounded (ovoid); venter bright red (cream). From *A. moisesii* sp. nov. by shorter hand, HAL/SVL 0.20–0.21, $n = 30$ (vs. 0.23–0.25, $n = 28$); venter bright red (pale yellow).

*Description of the holotype.* Body slender, elongate. Head triangular in lateral view and rounded in dorsal view. Head longer than wide. HL 39.6% of SVL. HW 31.3% of SVL. Snout acute in lateral view and pointed in dorsal view. SL 42.8% of HL. Nostrils slightly protuberant, closer to snout than to eyes. *Canthus rostralis* straight in dorsal view. Internarial distance smaller than eye diameter. IND about 27.6% of HW. Upper eyelid covered with tiny tubercles. Eye wide, prominent, about 30.3% of HL. Tympanum not visible through the skin. Skin around tympanum covered by tiny granules. Vocal sac not visible. Texture of dorsal skin finely granular. Texture of dorsolateral skin finely granular. Forelimbs slender. Edges of forelimbs with granules, in dorsal and ventral view. Upper arms robust. UAL 30.5% of SVL. Small granules from the outer edge of the mouth to upper arm. HAL 72.4% of UAL. Fingers basally webbed. Fingers slender, tips unexpanded. Relative length of fingers: I<II<IV<III. Supernumerary tubercles and accessory palmar tubercles rounded. Palmar tubercle large and rounded. Subarticular tubercles rounded. Texture of gular region finely granular. Texture of ventral skin granular. Small granules on venter. Hindlimbs slender. Edges of thigh to tarsus covered by conical tubercles. THL 51.8% of SVL. TAL 50.6% of SVL. Tarsus slender. TL 29.8% of SVL. FL 60% of THL. Relative length of toes: I<II<V<III<V. Inner metatarsal tubercle oval. Outer metatarsal tubercles small and rounded. Subarticular tubercles rounded. Toes slender and elongate. Tip of toes not expanded, unwebbed. Cloacal opening slightly above midlevel of thighs.

*Measurement of the holotype* (*in mm*). SVL 12.6; HW 3.9; HL 5.0; SL 2.1; ED 1.2; IND 1.1; UAL 3.8; HAL 2.7; THL 7.2; TAL 6.9; TL 3.9; FL 4.3.

*Variation* (Fig. 13). The new species presents extensive variation among specimens (e.g., https://bioweb.bio/galeria/FotosEspecimenes/Amazophrynella%20minuta/1). Sexual dimorphism was observed in SVL, with 11.5–14.7 mm (13.0 ± 0.6 mm, $n = 29$) in males and 16.1–20.8 mm (18.3 ± 0.9 mm, $n = 35$) in females. Specimens (MUBI 7686, MUBI 7698, MUBI 7699, MUBI 7700) from Andoas, Peru, present fewer tubercles on upper arm. Abundance of granules on ventral surfaces varies in density (e.g., QCAZ 21425, QCAZ 21431, QCAZ 20504, QCAZ 10526, QCAZ 46430). Some individuals (e.g., QCAZ 37761, QCAZ 6095, QCAZ 6105) present one to two large tubercles on dorsolateral region. Specimens from Pastaza (e.g., QCAZ 56342, QCAZ 56354, QCAZ 56361, QCAZ 38599, QCAZ 38679, QCAZ 38722) present greater abundance of granules on dorsum. Some individuals display different sized blotches on venter, while in other specimens, blotches are absent (Fig. 13C). In life, belly coloration varies between yellow to light red. The gular

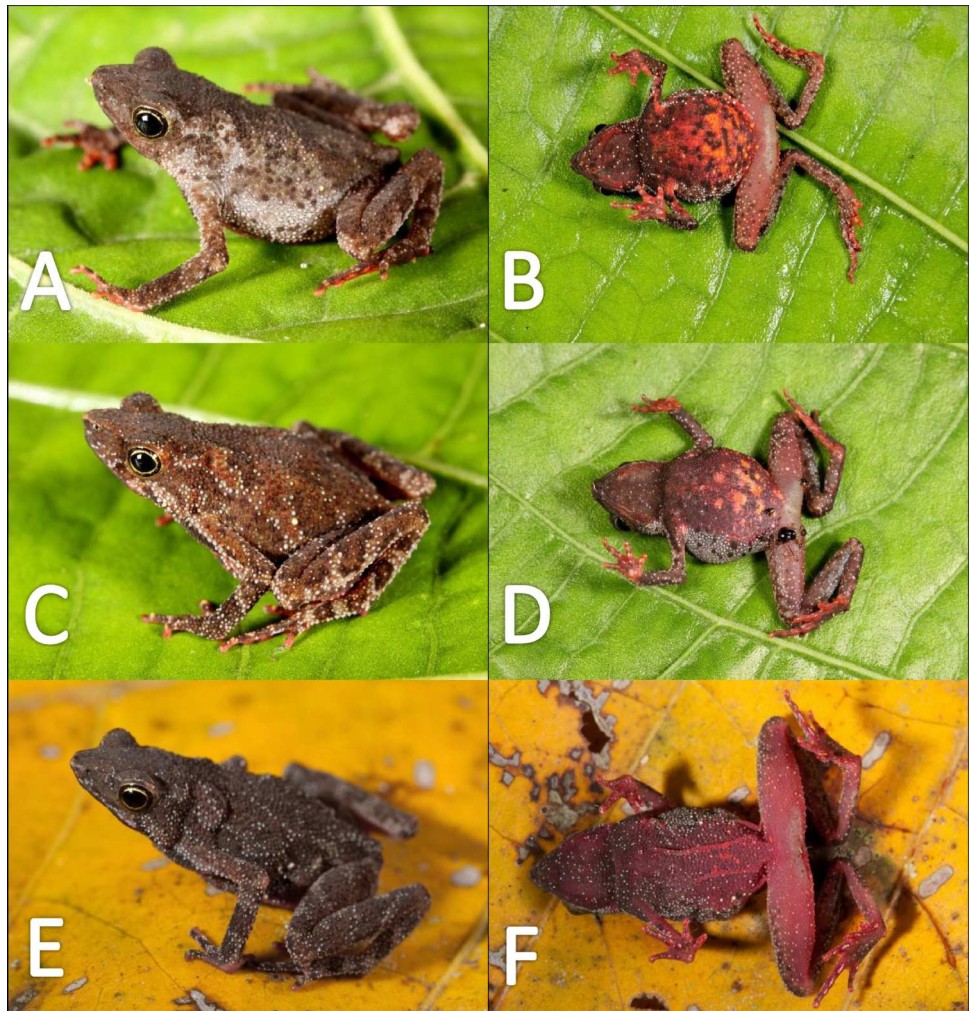

**Figure 13** **Morphological variations of live *Amazophrynella siona* sp. nov.** QCAZ 51068 (A–B); QCAZ 42988 (C–D); QCAZ 42988 (E–F). Photos by Santiago R. Ron.

region varies from light red to red. Thighs, shanks, tarsus and feet vary from light red to red, in dorsal view. Palm and sole color from light red to orange, in ventral view.

*Coloration of the holotype (in life)*. Head brown, in dorsal view. Dorsum mostly brown. Flanks reddish brown. Dorsal surfaces of upper arm, arm and hand light brown. Dorsal surfaces of the thighs, tibia, tarsus and foot light brown. Ventral surfaces of upper arm light red, arm light brown, palm reddish brown. Gular region reddish brown. Belly bright red with yellow blotches. Axillar region with yellow granules. Ventral surfaces of thighs, tarsus and tibia reddish brown, sole reddish brown. Iris golden and pupil black.

*Color in preservative* (∼70% ethanol, Fig. 14). Almost the same as color in life. Dorsum became brown. We detected a gradual fading of the red and yellow coloration of the chest and venter. The blotches on venter became less evident. Fingers and toes became pale red.

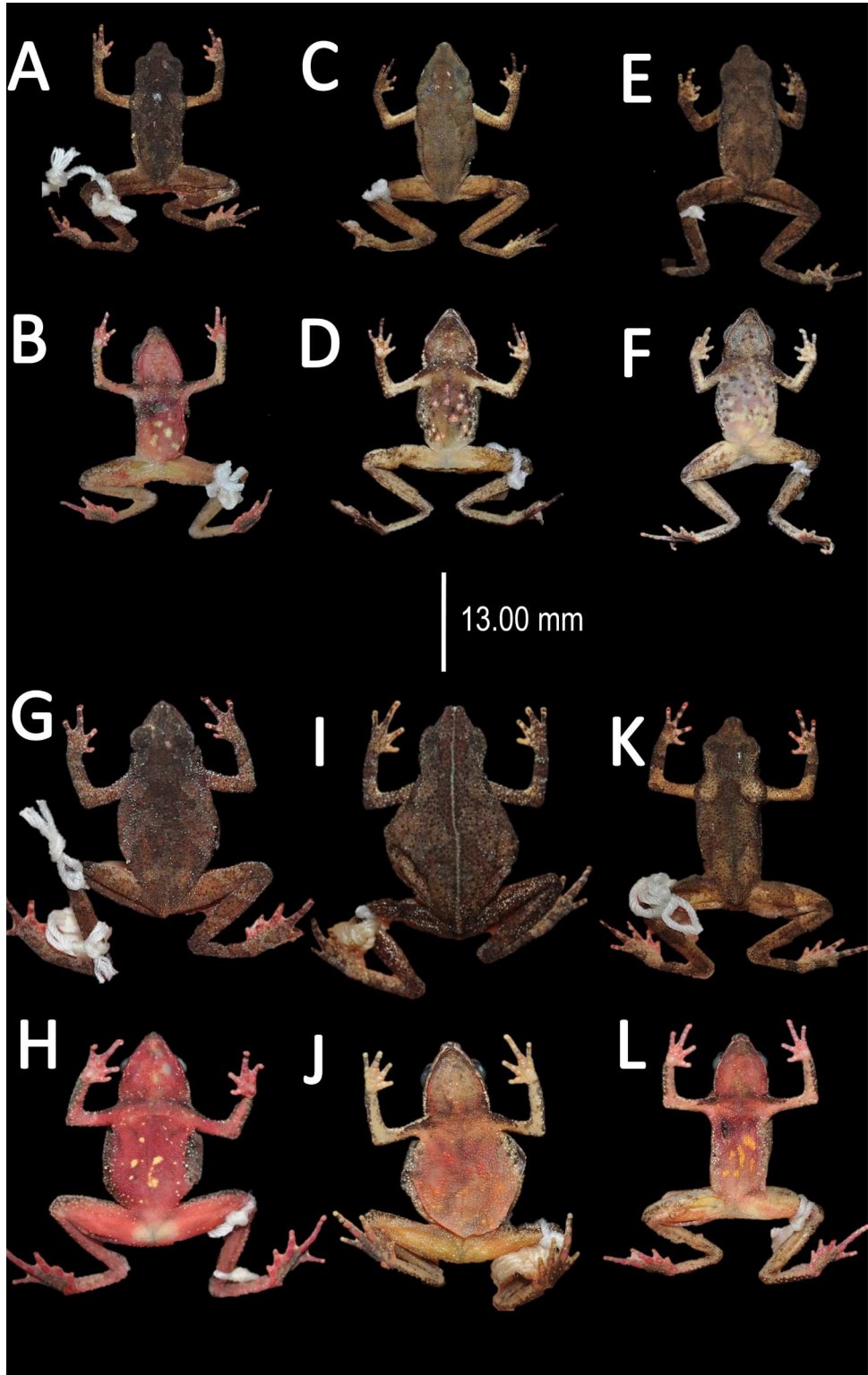

**Figure 14 Morphological variations of preserved specimens of *Amazophrynella siona* sp. nov.** Adult males: QCAZ 54213 (A–B); QCAZ 11979 (C–D); QCAZ 18826 (E–F). Adult females: QCAZ 38679 (G–H); QCAZ 6091 (I–J); QCAZ 52434 (K–L). Photos by Rommel R. Rojas.

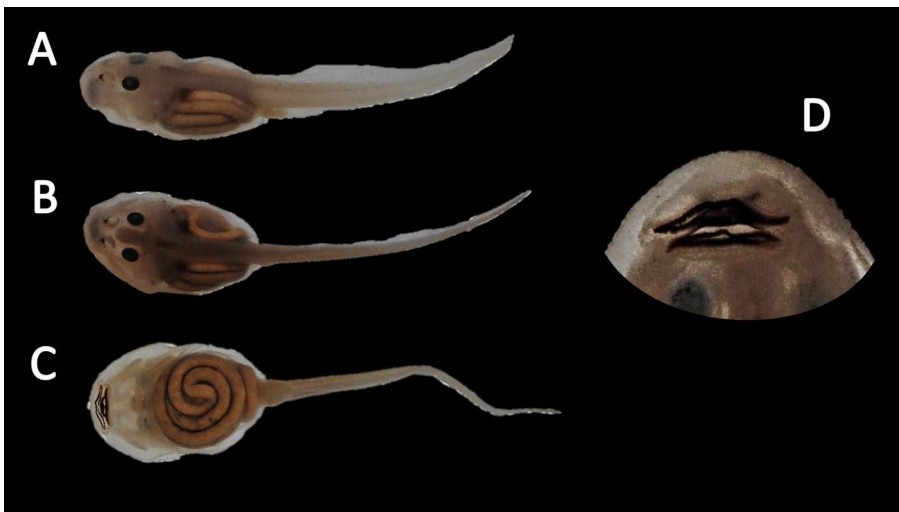

**Figure 15** **Tadpole of *Amazophrynella siona*. sp. nov.** National Park Yasuni, Ecuador (QCAZ 24576), stage 30; (A) dorsolateral view; (B) dorsal view; (C) ventral view; (D) oral disc view. Photos by Rommel R. Rojas.

*Tadpoles* (Fig. 15). *Duellman & Lynch (1969)* described the tadpole of *Amazophrynella siona* sp. nov. as *Atelopus minutus* based on ten individuals at stage 31 and three at stage 40, from Sarayacu, Province of Pastaza, 400 m a.s.l. The morphological characteristics described by *Duellman & Lynch (1969)* are similar to those observed by us. We analyzed ten tadpoles at stage 30. Body ovoid in dorsal view. Total length 11.0–13.2 mm (11.5 ± 0.84 mm). Body length 3.6–4.8 mm (4.2 ± 0.3 mm); depressed in lateral view. Body height 1.2–1.9 mm (1.5 ± 0.2 mm), body widest posteriorly. Snout rounded in dorsal and lateral view. Eye diameter 0.3–0.5 mm (0.3 ± 0.1 mm). Eye snout distance 0.9–1.4 mm (1.2 ± 0.14 mm). Nostrils small, closer to eyes than to tip of snout. Inter nasal distance 0.5–0.75 mm (0.6 ± 0.1 mm). Inter orbital distance 0.5–0.75 mm (0.6 ± 0.09 mm). Spiracle opening single, sinistral and conical. Spiracle opening on the posterior third of the body. Centripetal wall fused with the body wall and longer than the external wall. Upper and lower lips bare, single row of small blunt teeth, sectorial disc absent. Jaw sheaths finely serrated. Two upper and three lower rows of teeth. Oral disc weight 0.8–1.1 mm (0.9 ± 0.1 mm). Dorsal fin originating on the tail-body junction, increasing in height throughout the first third of the tail and decreasing gradually in the posterior two thirds of the tail to a pointed tip, in lateral view. Ventral fin originating at the posteroventral end of the body, higher at the first third of the tail, decreasing gradually in height toward tail tip. Tail length 5.4–8.1 mm (6.8 ± 0.9 mm). Tail height 0.9–1.1 mm (0.9 ± 0.1 mm). Body and tail rosaceous with small dark pointed flecks on body in fixed specimens. In life, *Duellman & Lynch (1969)* reported brown body and spotted tail with black and small brown flecks on caudal musculature, the entire dorsal fin and posterior third of ventral fin.

*Bioacoustics* (Fig. 16). The advertisement call of *Amazophrynella siona* sp. nov. was described by *Duellman (1978)* as the advertisement call of *Dendrophryniscus minutus* from Santa Cecilia, Ecuador. We analyzed one call from the Reserva de

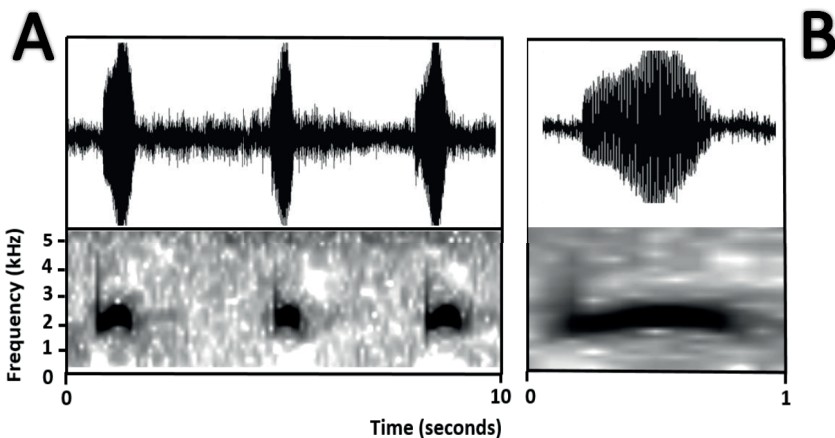

**Figure 16** Oscillogram and spectrogram of the advertisement call of *Amazophrynella siona* sp. nov. (A) Three notes; (B) one note.

Producción Faunistica Cuyabeno, Province of Sucumbíos, Ecuador (QCAZ 18833) (http://bioweb.puce.edu.ec/QCAZ/inicio). The call was recorded one day after capture, on February 6, 2002. In our analysis all the call parameters from *Duellman (1978)* overlap with the call of the new species. Call trill emitted at irregular intervals. Note duration 0.03–0.06 s (0.013 ± 0.001 s, $n = 16$). The fundamental frequency 2,000–3,240.1 Hz (3,000.9 ± 101.79 s, $n = 16$). Dominant frequency 3,647.5–4,200 Hz (3,757.9 ± 138.1 Hz, $n = 16$). The number of pulses 23–28 pulses per note (28.5 ± 5.3 pulses/note, $n = 16$). Time to peak amplitude 0.01–0.03 s (0.02 ± 0.01 s, $n = 13$). The call has a downward modulation, reaching its maximum frequency almost at the middle.

*Distribution and natural history* (Fig. 1B). *Amazophrynella siona* sp. nov. have been recorded from Ecuador, in Provinces of Orellana, Sucumbíos and Pastaza and Peru in the Province Andoas, northern Loreto Department. It occurs at elevations ranging from 200–900 m a.s.l. The species is found in the leaf litter of primary and secondary forest, terra firme or flooded forest, and swamps. It is active during the day; at night individuals rest on leaves, usually less than 50 cm above ground. It breeds throughout the year (*Duellman, 1978*). This species shows conspicuous sexual dimorphism, with males being much smaller than females. The amplexus is axillar. Eggs are pigmented; males call from amidst leaf litter. *Duellman & Lynch (1969)* reported that this species deposited its eggs in gelatinous strands 245–285 mm long, with 245–291 eggs. It can be abundant at some sites (e.g., Cuyabeno reserve; SR Ron, pers. obs., 2018) Given its large distribution range (>20,000 km$^2$) which also includes vast protected areas and locally abundant populations, we suggest assignment this species to the Least Concern category.

*Etymology.* The specific epithet is a noun in apposition and refers to the Siona, a western Tucanoan indigenous group that inhabits the Colombian and Ecuadorian Amazon. The Siona inhabit the Cuyabeno Lakes region, an area where *Amazophrynella siona* sp. nov. is be abundant. While working in his undergraduate thesis in the early 1990s, SRR lived with the Siona at Cuyabeno. The Siona chief, Victoriano Criollo, had an encyclopedic knowledge

of the natural history of the Amazonian forest, superior in extent and detail to that of experienced biologists. His death, a few years ago, represents one of many instances of irreplaceable loss of traditional knowledge triggered by cultural change among Amazonian Amerindians.

*Amazophrynella xinguensis* sp. nov.
urn:lsid:zoobank.org:act:55CD4C19-9A39-4DEB-BA6C-F02F9735BB77

*Amazophrynella* cf. *bokermanni* (Vaz–Silva et al. 2015: 208, "Volta grande", Xingu River, Pará, Brazil)

*Holotype* (Fig. 17). INPA–H 35471, adult male, collected at the Sustainable Development Project (PDS) Virola Jatobá (3°10′06″S, 51°17′54.2″W), 86 m a.s.l., municipality of Anapú, state of Pará, Brazil by E. Hernández and E. Oliveira on December 06, 2012.

*Paratypes.* Twenty two specimens (males = 4, females = 14, immatures = 4). Brazil: Pará State: Municipality of Senador José Porfírio: Fazenda Paraíso (2°34′37″S, 51°49′50.3″W), 57 m a.s.l., INPA–H 35482, INPA–H 35493 (adult males), INPA–H 35472 (adult female), E. Hernández and E. Oliveira on December 05, 2012. Municipality of Anapu: PDS Virola Jatobá, (3°10′06″S, 51°17′54.2″W), 86 m a.s.l., INPA–H 35484, INPA–H 35485 (adult males), INPA–H 35473, INPA–H 35474, INPA–H 35475, INPA–H 35476, INPA–H 35477, INPA–H 35478, INPA–H 35479, INPA–H 354780, INPA–H 35481, INPA–H 35483, INPA–H 35490, INPA–H 35491, INPA–H 3592 (adult females), E. Hernández and E. Oliveira on December 06, 2012. Municipality of Vitória do Xingu, Ramal dos Cocos (3°09′42.1″S, 52°07′41.9″W), 110 m a.s.l., INPA–H 35486, INPA–H 35487, INPA–H 3588, INPA–H 35489 (immatures), E. Hernández and E. Oliveira on December 04, 2012.

*Diagnosis.* An *Amazophrynella* with (1) SVL 17.0–20.0 mm in males, 22.4–26.3 mm in females; (2) snout pointed in lateral view; (3) upper jaw, in lateral view, protruding beyond lower jaw; 4) tympanums, vocal sac, parotid gland and cranial crest not evident; (5) texture of dorsal skin highly granular; (6) abundance of small tubercles on dorsum, on upper arm and on arms; (7) texture of ventral skin granular; (8) fingers I and II basally webbed; (9) finger III relative short (HAL/SVL = 0.20–0.22, $n = 18$); (10) thumb larger and robust; (11) finger I larger or equal than finger II, FI = 2.1 vs. FII = 2.1 in adult males, $n = 5$ and FI = 2.8 mm, vs. FII = 2.9 mm, in adult females, $n = 13$; (12) palmar tubercle ovoid; (13) toes slender, basally webbed; in life: (14) venter greyish; black dots on venter.

*Comparison with other species (characteristics of compared species in parentheses).* *Amazophrynella xinguensis* sp. nov. is more similar to *A. bokermanni* from which it can be distinguished by: texture of dorsal skin highly granular (granular); relative size of fingers: FI ≥ FII mean 2.1 mm, in I vs. 2.1 mm in II in *A. xinguensis* sp. nov. $n = 5$ (vs. FI >FII, mean 2.2 mm in FI vs. in 2.0 mm FII in *A. bokermanni*, $n = 7$, Figs. 6C vs. 6D; shape of palmar tubercle elliptical (rounded); presence of tubercles on dorsum (absent); dorsal coloration dark brown (light brown); venter light gray (white); gular region dark brown (grayish brown). From the other species of *Amazophrynella* the new species is easily differentiated by having FI ≥ FII (FI <FII in all the other species, Fig. 6); its greater SVL of males (KW $x_2 = 108.6$, $df = 10$, $p$-value = 0.001, Fig. 5) and its protruding ovoid palmar tubercle (vs.

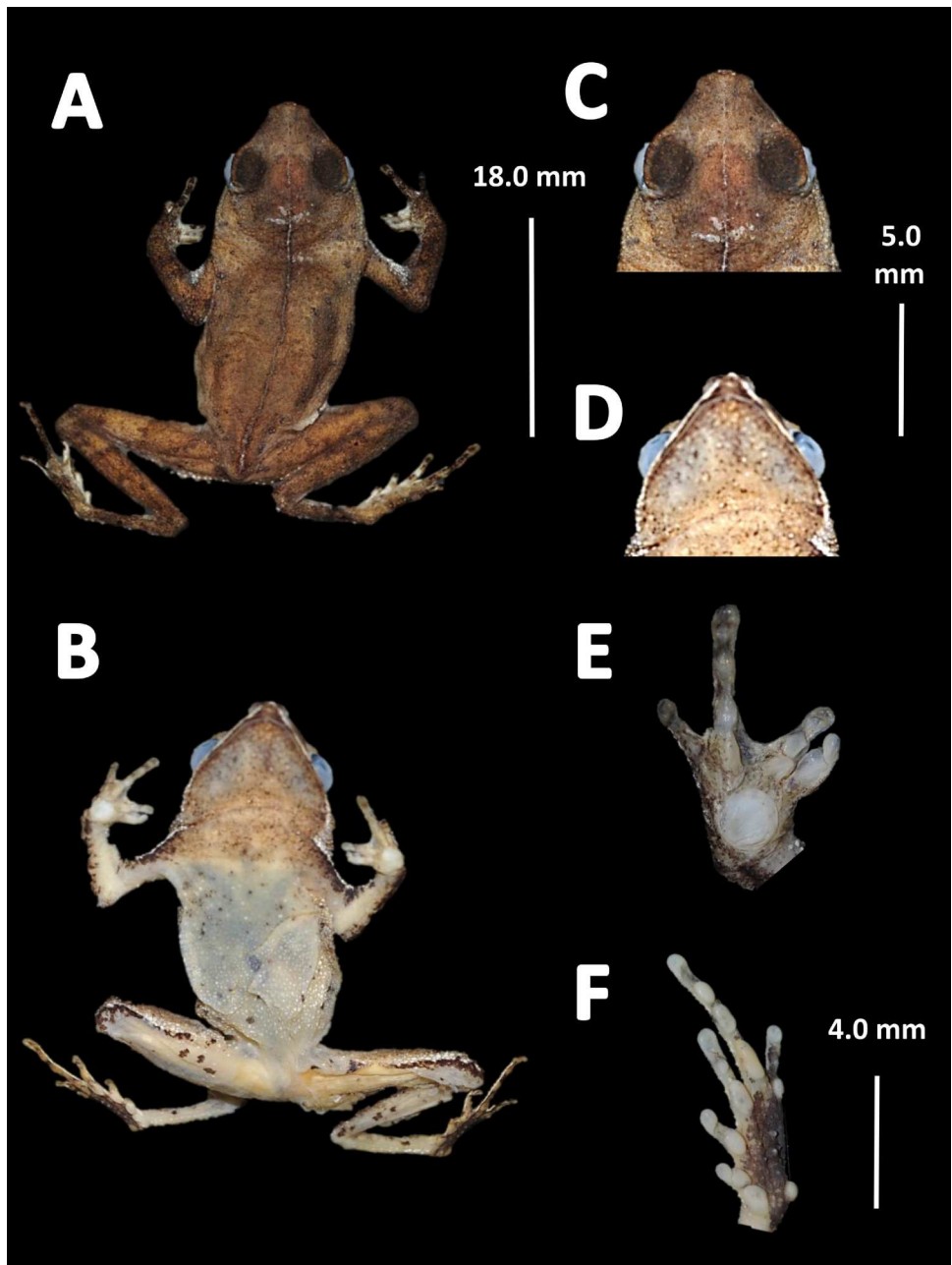

**Figure 17 Holotype of *Amazophrynella xinguensis*. sp. nov. (INPA-H 35471).** (A) Dorsal view; (B) ventral view; (C) ventral view of head; (D) dorsal view of head; (E) right hand; (F) right foot. Photos by Rommel R. Rojas.

*A. teko, A. manaos, A. vote, A. minuta, A. bokermannni, A. javierbustamantei, A. matses, A. Amazonicola, A. siona* sp. nov. *A. teko* sp. nov., *A. moisesii* sp. nov. see Fig. 6).

*Description of the holotype.* Body robust. Elongate. Head pointed in lateral view and triangular in dorsal view. Head longer than wide. HL 35.5% of SVL. HW 27.1% of SVL. Snout acute in lateral view and triangular in dorsal and ventral view. SL 64.0% of HL.

Nostrils slightly protuberant, closer to snout than to eyes. *Canthus rostralis* straight in dorsal view. Internarial distance smaller than eye diameter. IND about 20.8% of HW. Upper eyelid covered by small granules. Eye prominent, 30.3% of HL. Tympanum not visible through the skin. Skin around tympanum covered by tiny granules. Vocal sac not visible. Texture of dorsal skin highly granular. Rounded small tubercles on dorsum. Texture of dorsolateral skin granular. Forelimbs thick. Edges of arms of forelimbs with granules, in dorsal and ventral view. Upper arms robust. UAL 28.5% of SVL. Abundance of small tubercles on upper arm. HAL 68.4% of UAL. Fingers slender, tips unexpanded. Fingers basally webbed on finger II and finger III. Relative length of fingers: I ≥II<IV<III. Supernumerary tubercles rounded. Palmar tubercle ovoid. Gular region finely granular. Texture of ventral skin granular. Small granules in the venter. Hind limbs slender. Edges of thigh to tarsus covered by conical tubercles. THL 52.2% of SVL. Tibias almost the same length as thighs. TAL 48.9% of SVL. Tarsus slender. TL 29.8% of SVL. FL 60.0% of THL. Relative length of toes: I<II<III<V<IV. Inner metatarsal tubercle oval. Outer metatarsal tubercles small and rounded. Subarticular tubercles rounded. Toes slender. Tip of toes not expanded, basally webbed. Cloacal opening slightly above midlevel of thighs.

*Measurement of the holotype* (*in mm*). SVL 18.5, HW 5.0, HL 6.0, SL 3.1, ED 2.1, IND 1.6; UAL 6.6; HAL 4.1, FI 1.9, FII 1.9, THL 9.7, TAL 9.3, TL 5.7, FL 6.4.

*Variation* (Fig. 18). Sexual dimorphism was observed in SVL, with 17.7–20.0 mm (18.9 ± 1.0 mm, $n = 5$) in males and 22.4–26.3 mm (24.1 ± 1.2 mm, $n = 13$) in females. Some individuals (i.e., INPA–H 35473, INPA–H 35477, INPA–H 35475) present one to two large tubercles on dorsolateral region. The granules on ventral surfaces are greatly abundant in some individuals (e.g., INPA–H 35478, INPA–H 35480, INPA–H 35486). The gular region presents black or brown coloration. Dots on venter display different sizes (small to medium) and abundance (Figs. 18D vs 18A. In life, ventral surfaces from cream to light gray. Thighs, shanks and tarsus between cream to white coloration, in ventral view. Palm and sole present different tonalities of orange, in ventral view.

*Coloration of the holotype (in life).* Head dark brown, in dorsal view. Dorsum mostly light brown with brown chevrons. Flanks cream. Dorsal surfaces of upper arm, arm and hand light brown. Dorsal surfaces of thighs, tibia, tarsus and foot brown. Ventral surfaces of upper arm, arm and palm cream. Ventral surfaces of thighs, tarsus and tibia cream, sole black. Gular region cream. Belly cream with tiny black blotches. White line from the tip of snout to cloaca. Iris golden and pupil black.

*Color in preservative* (∼70% ethanol, Fig. 19). In preservative, the coloration is almost the same than life. The coloration of the dorsum became dark brown. Gular region and venter became white. The iris loses its coloration. The fingers and toes became cream.

*Distribution and natural history* (Fig. 1B). *Amazophrynella xinguensis* sp. nov. have been recorded from State of Pará, Brazil, at three localities: PDS Virola Jatoba, municipality of Anapú, Fazenda Paraiso, municipality of Senador José Porfirio (right bank of Xingu River) and Ramal dos Cocos, municipality of Altamira (left bank of Xingu River), all of them in area of influence of the Belo Monte dam. It occurs in elevations of 86–106 m a.s.l. This species is found amidst leaf litter. The amplexus is axillar (Fig. 18C). Reproduction occurs in the rainy season in tiny puddles. Males were found hidden in the leaf litter. Tadpoles and

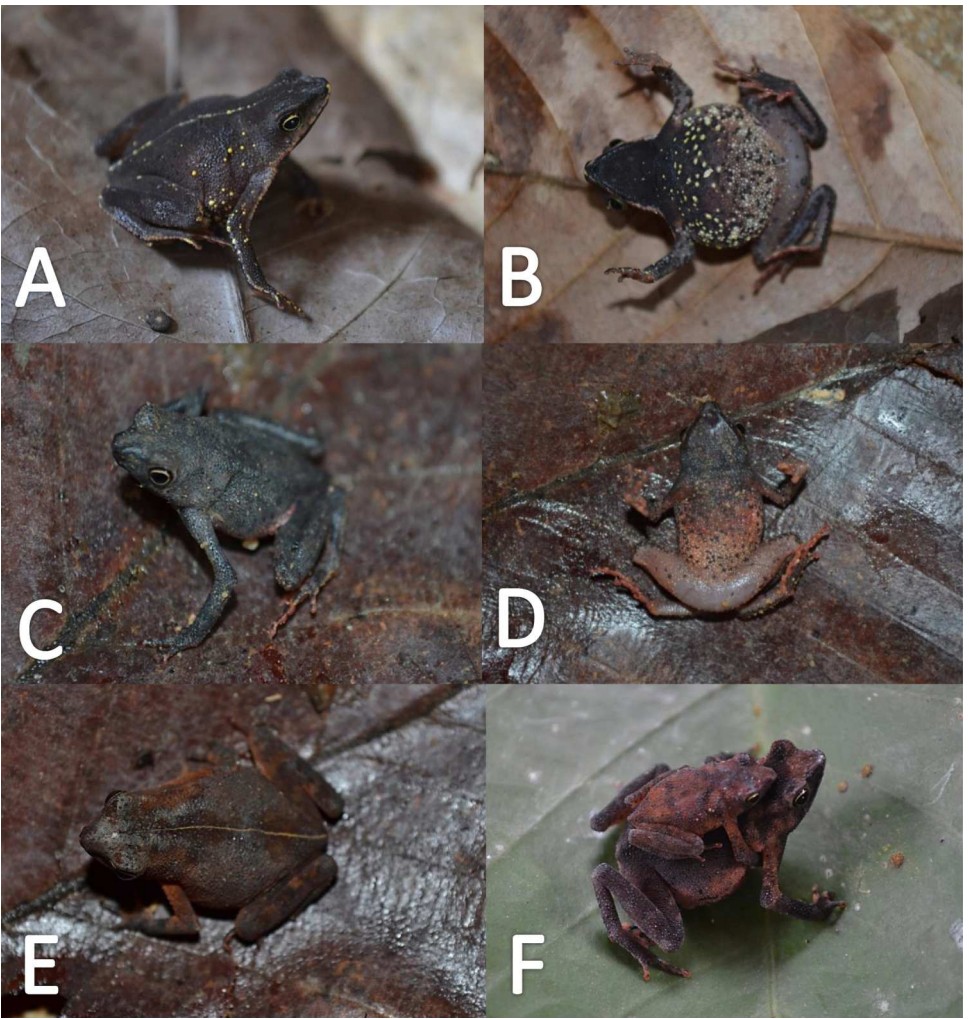

**Figure 18** **Morphological variation of live *Amazophrynella xinguensis* sp. nov. (unvouchered specimens).** Adult females (A–D); dorsal variation (E); Amplexus (F). Photos by Emil Hernández-Ruz.

advertisement call are unknown. The conservation status of this species remains unknown, but the recent construction of the Belo Monte hydroelectric complex on the Xingu River represents a threat to the population status of this species.

*Etymology.* The specific epithet refers to geographic distribution of the species within the lower Xingu River basin, Brazil.

*Amazophrynella moisesii* sp. nov.
urn:lsid:zoobank.org:act:9984F3CB-9416-482D-8F63-5D78C8CDC032
*Dendrophryniscus minutus* (Bernarde et al. 2011: 120 plate 2, Fig. d)
*Amazophrynella minuta* (Bernarde et al. 2013: 224, 227 plate 7 Fig. c; Miranda et al. 2015: 96)

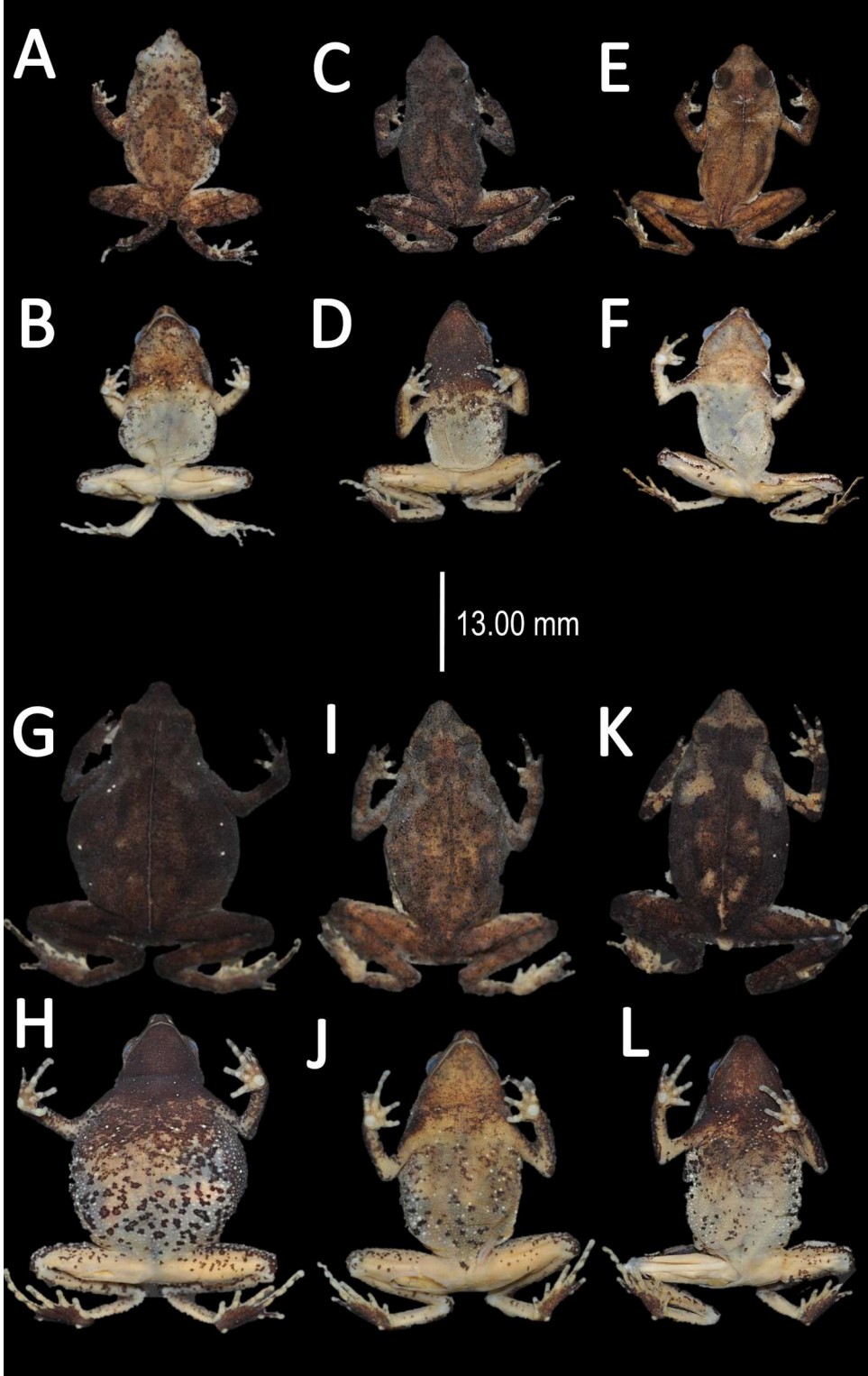

**Figure 19 Morphological variation of preserved specimens of *Amazophrynella xinguensis* sp. nov.**
Adult males: INPA-H 35482 (A–B), INPA-H 35493 (C–D); INPA-H 35471 (E–F). Adult females: INPA-H 35477 (G–H); INPA-H 35478 (I–J); INPA-H 35479 (K–L). Photos by Rommel R. Rojas.

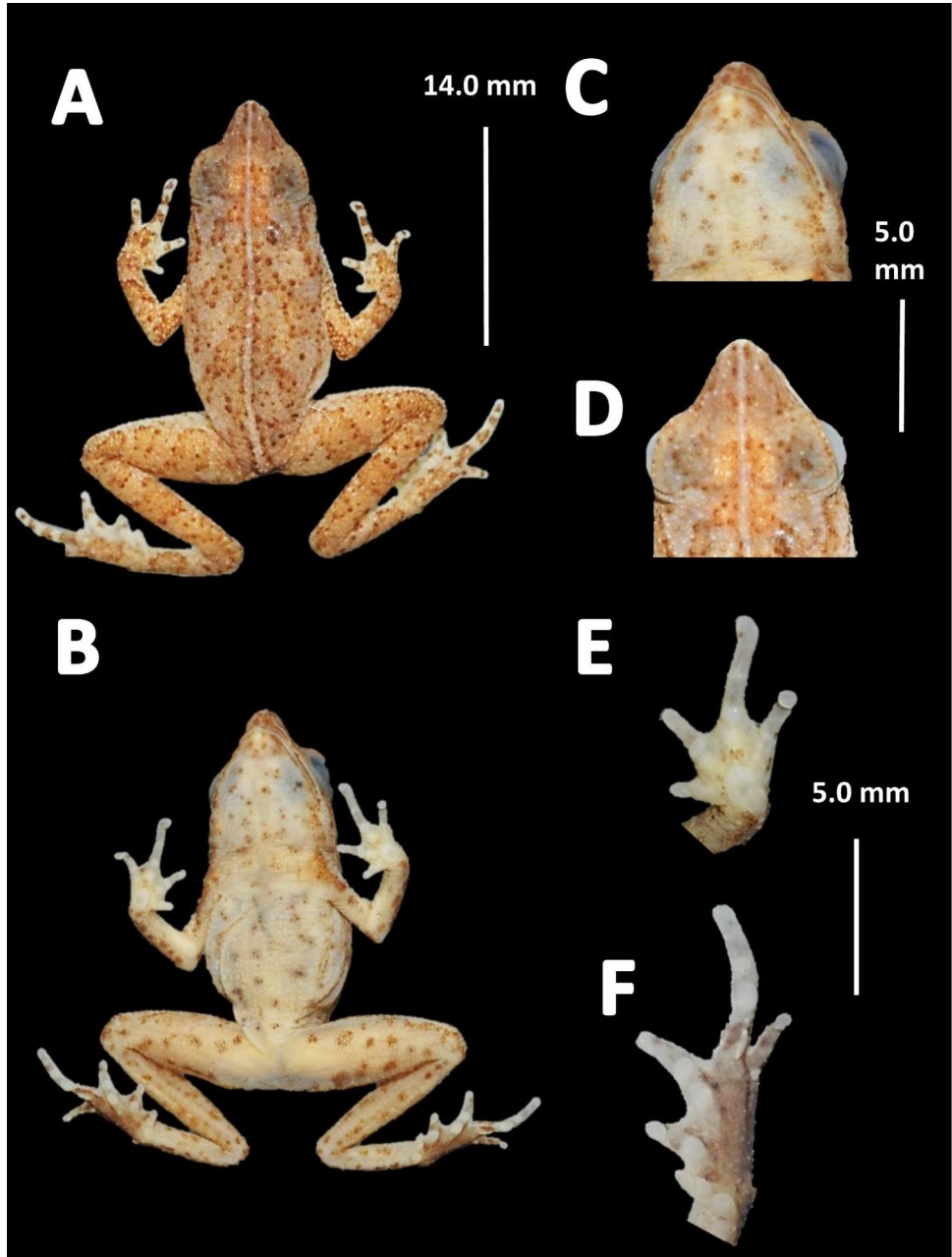

**Figure 20 Holotype of *Amazophrynella moisesii*. sp. nov. (UFAC-RB 2815).** (A) Dorsal view; (B) ventral view; (C) ventral view of head; (D) dorsal view of head; (E) right hand; (F) right foot. Photos by Rommel R. Rojas.

*Holotype* (Fig. 20). UFAC–RB 2815 adult male, collected in the Parque Nacional da Serra do Divisor, Igarapé Ramon (7°27′00″S, 73°45′00″W), 400 m a.s.l., municipality of Mâncio Lima, Acre, Brazil by Moises Barbosa de Souza on 1 January, 2000.

*Paratypes.* Thirty eight specimens (males = 18, females = 20), Acre, Brazil: Reserva Extrativista Alto do Juruá (9°03′00″S, 72°17′00″W), 260 m a.s.l., UFAC–RB 823 (adult

male), Moisés B. Souza and Adão J. Cardoso on 26 February 1994, UFAC–RB 878–879 (adult males), Moisés B. Souza and Paulo Roberto Manzani between 16 and 18 July 1994; UFAC–RB 2606–2611 (adult females), Moisés B. Souza and M. Nascimento between 7 and 8 March 1998. Parque Nacional da Serra do Divisor: Igarapé Anil (8°59′00″S, 72°29′00″W), 192 m a.s.l., UFAC–RB 1337–1341 (adult females), UFAC–RB 1343 (adult female), Moisés B. Souza and William Aiache on 10 November 1994; Zé Luiz lake (8°54′00″S, 72°32′00″W), UFAC-RB 1774–1775 (adult females), Moisés B. Souza and William Aiache between 9 and 10 November 1996; Igarapé Ramon (7°27′00″S, 73°45′00″W), 400 m a.s.l., UFAC–RB 1375 (adult female), Moisés B. Souza and William Aiache between 12 and 13 November 1996, UFAC–RB 2772–2773 (adult females), UFAC–RB 2816–2817 (adult males), Moisés B. Souza between 18 and 20 January 2000; Môa River (7°30′00″S, 73°36′00″W), 331 m a.s.l, UFAC–RB 1493 (adult male), Moisés B. Souza and William Aiache between 19 and 20 November 1997, UFAC–RB 2687–2697 (adult males), Moisés B. Souza on 10 January 2000. Floresta Estadual do Gregório, municipality of Tarauacá (7°59′00″S, 71°22′36.8″W), 240 m a.s.l., UFAC–RB 5678 (adult female), Moisés B. Souza and Marilene Vasconcelos between 23 and 26 July 2000; Centrinho do Aluísio site, municipality of Porto Walter UFAC–RB 6273 (adult male), Paulo Roberto Melo Sampaio, on 8 January 2014. Municipality of Mâncio Lima, Acre (7°23′10.32″S, 73°3′31.68″W), MNRJ 91670 (field number PRMS 420) (adult female) Paulo Roberto Melo Sampaio and Evan M. Twomey on 24 March 2016. Amazonas state: Municipality of Envira (7o31′16.14″S, 70o1′3.84″W), MNRJ 91669 (field number PRMS 404) (adult female) Paulo Roberto Melo Sampaio and Evan M. Twomey on 12 March 2016.

*Diagnosis.* An *Amazophrynella* with (1) SVL 12.2–15.8 mm in males, 16.4–20.9 mm in females; (2) snout acuminate in lateral view, upper jaw, in lateral view, protruding beyond lower jaw; (3) snout length protuberant, large for the genus (SL/HL = 0.48–0.5); (4) cranial crest, vocal slits and nuptial pads absent; (5) small tubercles on upper arms and posterior area of tympanums; (6) texture of dorsal skin tuberculate; (7) texture of ventral skin highly granular (8) finger III relative large (HAL/SVL 0.23–0.25, $n = 28$); (9) fingers slender, basally webbed; (10) finger I shorter than finger II; (11) palmar tubercle elliptic; (12) hind limbs relatively large (TAL/SVL 0.51–0.53, $n = 28$); (13) toes slender basally webbed; in life: (14) venter pale yellow; small irregular dots on venter.

*Comparison with other species (characteristics of compared species in parentheses).* *Amazophrynella moisesii* sp. nov. is most similar to *A. javierbustamantei* from which it can be distinguished by: protruding snout, SL/HL 0.48–0.5, $n = 28$ (vs. 0.43–0.45, $n = 60$); snout acuminate, in lateral view (subacuminate); ventral skin highly granular (coarsely areolate); larger hind limbs, TAL/SVL 0.51–0.53, $n = 28$ (vs. 0.49–0.51, $n = 60$); venter bright yellow (pale yellowish orange); small irregular blotches on venter (tiny rounded points). From the other species of the genus *Amazophrynella* the new species is easily differentiated by its large hand, HAL 3.6–5.6 mm (4.62 ± 0.62 mm) in adult females, 2.5–4.1 mm (3.4 ± 0.52 mm) in adult males (KW $x_2 = 100.2$, $df = 10$, $p$-value = 0.001, Fig. 21); longer SL, adult females 3.4–2.5 mm (3.0 ± 0.2 mm) and adult males 2.1–3.0 mm (2.6 ± 0.3 mm, KW $x_2 = 104.3$, $df = 10$, $p$-value = 0.001, Fig. 22); FI <FII (FI >FII in *A. bokermanni*, and FI ≥ FII in *A. xinguensis* sp.nov. - Figs. 6K vs. 6C and Figs. 6K vs. 6D and

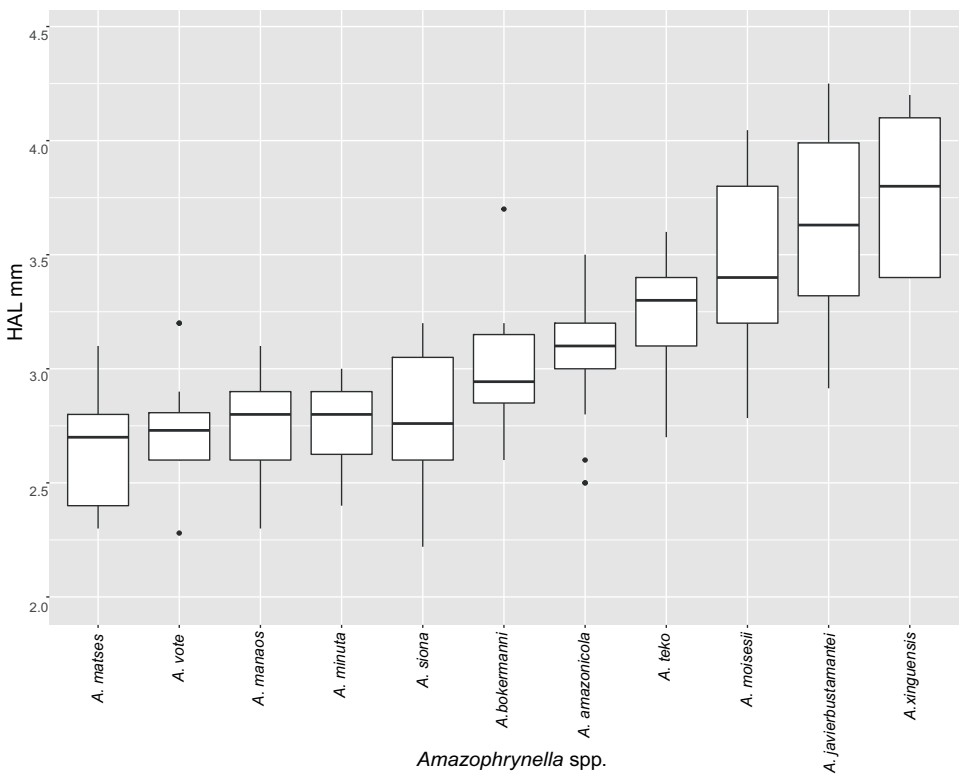

**Figure 21  Measurement comparison of HAL between males of nominal species of *Amazophrynella*.**

venter coloration pale yellow (white, in *A. manaos,* cream in *A. teko* sp. nov., red brown in *A. vote* and reddish brown in *A. siona* sp. nov., see Fig. 7).

*Description of the holotype.* Body slender, elongate. Head triangular in lateral view and pointed in dorsal view. Head longer than wide. HL 33.8% of SVL. HW 30.8% of SVL. Snout prominent, acuminate in lateral view and pointed in dorsal view. SL 50.9% of HL. Nostrils closer to snout than to eyes. *Canthus rostralis* straight in dorsal view. Internarial distance smaller than eye diameter. IND about 30.9% of HW. Upper eyelid covered by abundant granules on borders. Eye prominent, about 35.7% of HL. Tympanum not visible through the skin. Skin around tympanum covered by small granules. Vocal sac not visible. Texture of dorsal skin tuberculate. Abundance of granules on dorsum. Dorsolateral skin granular. Forelimbs slender. Edges of forelimbs covered by small conical granules, in dorsal and ventral view. Upper arms slender. UAL 35.2% of SVL. Small conical granules from the outer edge of the mouth to upper arm. Upper arm covered by abundant medium size granules. Large HAL. HAL 72.9% of UAL. Fingers basally webbed. Fingers slender, tips unexpanded. Relative length of fingers: I<II<IV<III. Supernumerary tubercles and accessory palmar tubercles rounded. Palmar tubercle large and elliptic. Subarticular tubercles rounded. Texture of gular region tuberculate. Texture of ventral skin highly granular. Small granules on venter. Hindlimbs slender. Thigh to tarsus covered by conical granules on borders. THL 54.4% of SVL. Tibias almost the same length as thighs. TAL

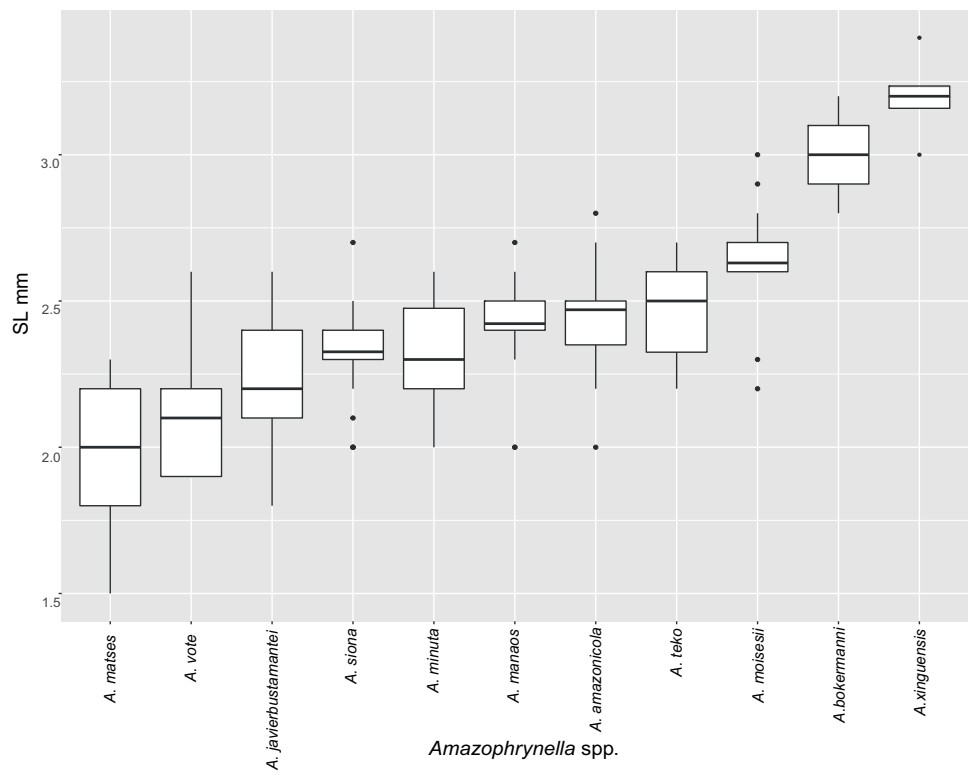

**Figure 22** Measurement comparison of SL between males of nominal species of *Amazophrynella*.

53.6% of SVL. Tarsus slender. TL 33.8% of SVL. FL 74.3% of THL. Relative length of toes: I<II<V<III<V. Inner metatarsal tubercle rounded. Outer metatarsal tubercles small and rounded. Subarticular tubercles rounded. Toes slender and elongate. Tip of toes not expanded, basally webbed. Cloacal opening slightly above middle of thighs.

*Measurement of the holotype* (*in mm*). SVL 13.6, HW 4.2, HL 5.1, SL 2.6, ED 1.5, IND 1.3; UAL 4.8; HAL 3.5, THL 7.4, TAL 7.3, TL 4.5, FL 5.5.

*Variation* (Fig. 23). Phenotypically, the new species present some variation among specimens. Sexual dimorphism was observed in SVL, with 12.2–15.8 mm (14.3 ± 1.5 mm, $n = 15$) in males and 16.4–20.9 mm (18.5 ± 1.6 mm, $n = 15$) in females. Some specimens present greater abundance of granules on dorsum (e.g., UFAC–RB 2690). Some individuals present greater abundance of small tubercles on dorsolateral region (e.g., UFAC–RB 2611, UFAC–RB 2603, UFAC–RB 2689, UFAC–RB 2692). Another specimen (UFAC–RB 2610) presents brown chevrons extending from the head to the vent, in dorsal view. Some individuals (e.g., UFAC–RB 829) present a line on dorsum, extending from the tip of the snout to cloaca. The pale yellow coloration of ventral surfaces may extend from thighs to the chest or just to the middle of the venter. In some specimens, the irregular black dots on venter vary in abundance and size (e.g., Figs. 24B vs. 24E). In life and preserved specimens, venter coloration between pale yellow and yellow. In some

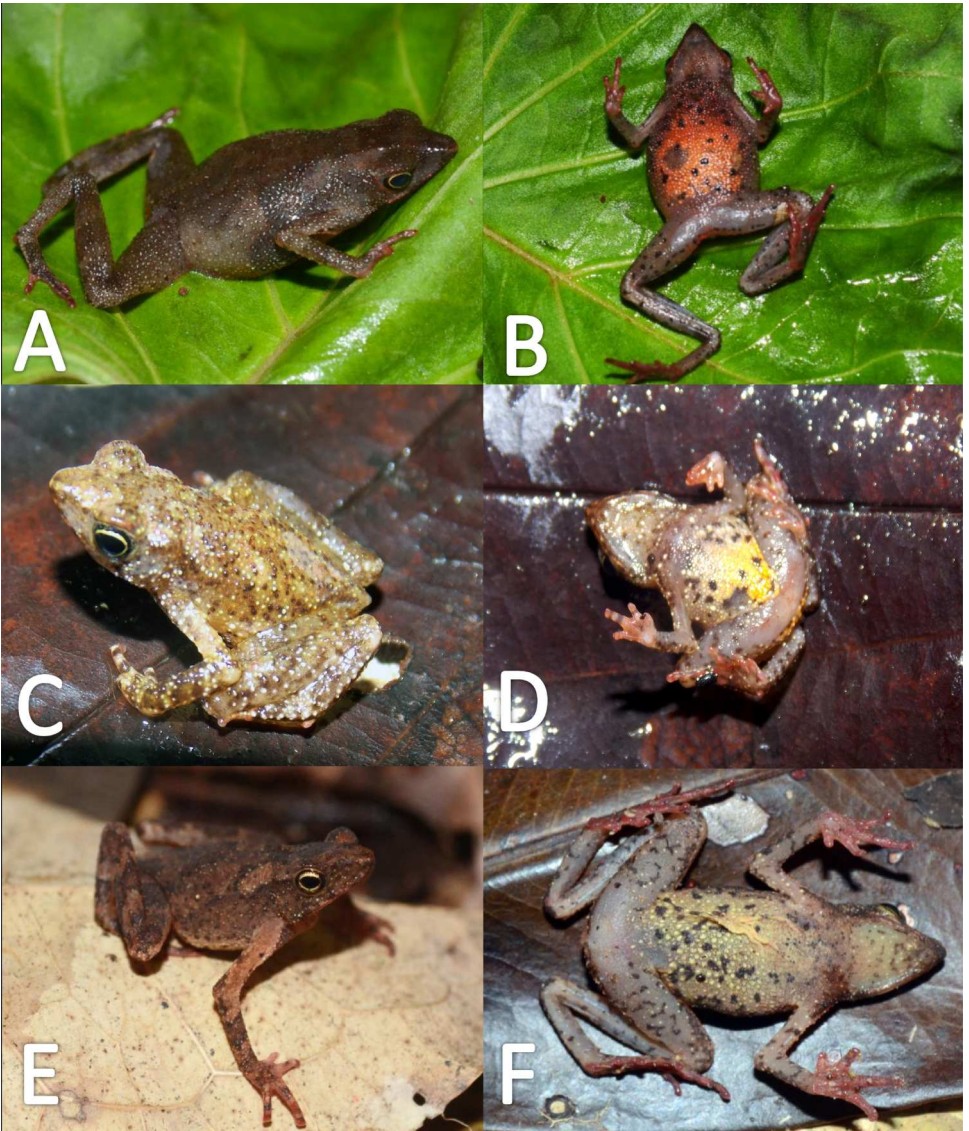

**Figure 23** **Morphological variation in live *Amazophrynella moisesii* sp. nov. (unvouchered specimens).** Adult females (A–B, E–F); adult males (C–D). Photos by Paulo R. Melo-Sampaio.

individuals, the thighs are abundantly covered by rounded tiny spots extending to the shank (Figs. 24C vs. 24D).

*Coloration of the holotype (in life).* Head brown, in dorsal view. Dorsum mostly light brown. Flanks cream with scattered small black dots. Dorsal surfaces of upper arm, arm and hand light brown. Dorsal surfaces of thighs, tarsus and foot light brown. Ventral surfaces of upper arm, arm and palm cream. Ventral surfaces of thighs, tarsus and tibia cream with small black dots. Sole light brown. Fingers cream, in ventral view. Gular region cream with small dots. Venter pale yellow with small dots. Iris golden and pupil black.

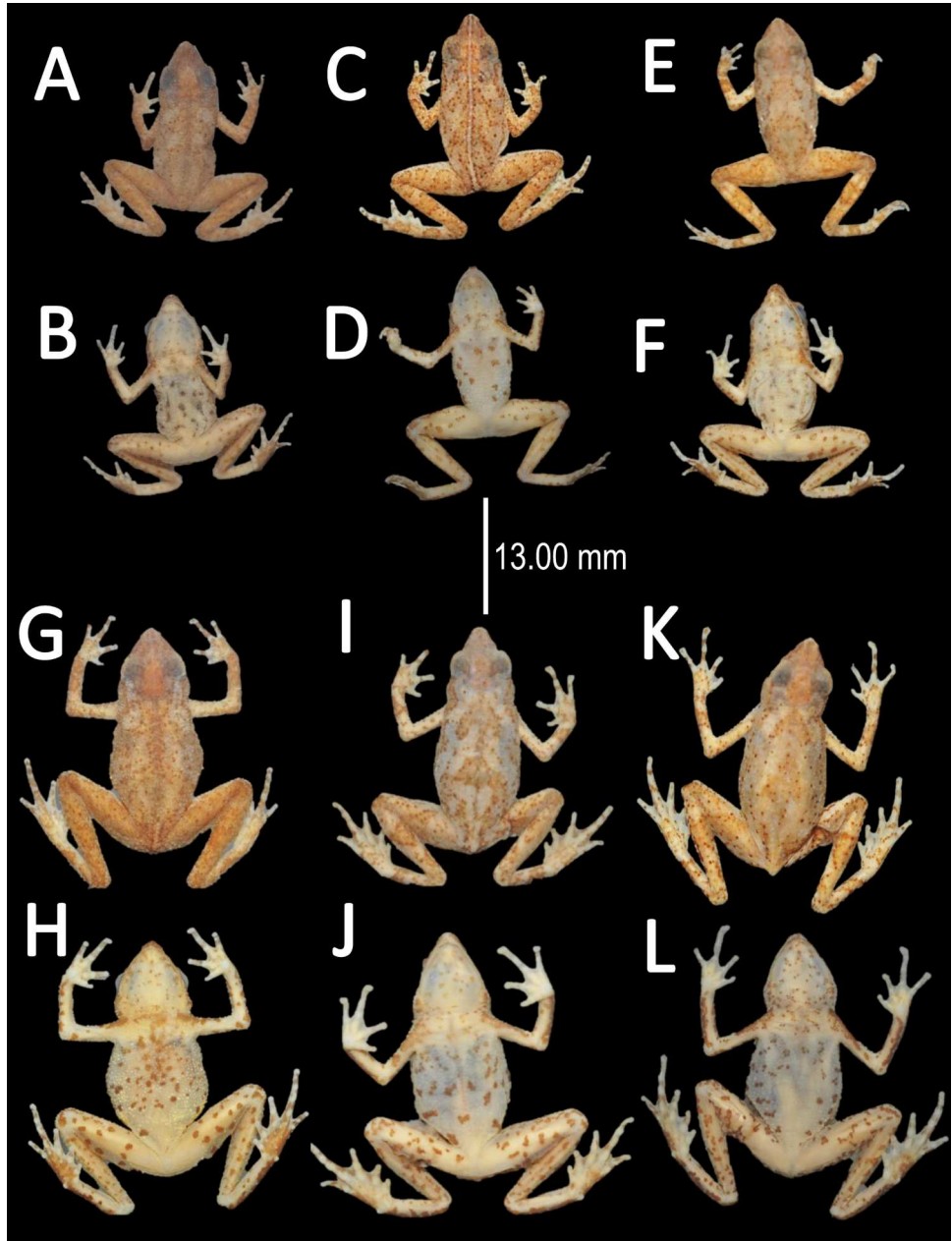

**Figure 24** **Morphological variations of preserved specimens of *Amazophrynella moisesii* sp. nov.** Adult males: UFAC-RB 1698 (A–B); UFAC-RB 2694 (C–D); UFAC-RB 2815 (E–F). Adult females: UFAC-RB 2608 (G–H); UFAC-RB 2610 (I–J); UFAC-RB 2607 (K–L). Photos by Rommel R. Rojas.

*Color in preservative* (~70% ethanol, Fig. 24). Nearly the same as color in life. The dorsum became light brown. We detected a fading of pale coloration of the chest and venter became cream. The small irregular dots on venter became less evident. The hand and foot became cream, in ventral view. The gular region and venter became cream. The iris loses its coloration.

*Distribution and natural history* (Fig. 1B). *Amazophrynella moisesii* sp. nov. have been recorded from Brasil. State of Acre: municipalities of Cruzeiro do Sul, Mâncio Lima, Porto Walter and Tarauacá; State of Amazonas: municipality of Envira. Peru: Department of Huanuco, Panguana, Rio Llullapichis. Due to its abundance and presence in conservation units of Brazil (Floresta Estadual do Gregório, Reserva Extrativista do Alto Juruá and Parque Nacional da Serra do Divisor) we recommend the IUCN Least Concern category.

*Etymology.* The specific epithet refers to Dr. Moisés Barbosa de Souza, a Brazilian biologist, professor and friend at the Universidade Federal do Acre (UFAC), to whom we dedicate this species in recognition of his contributions to herpetological research and amphibian conservation in the state of Acre, Brazil.

## DISCUSSION

To date, no study that analyzed a broadly distributed Amazonian taxon confirmed the existence of just one broadly distributed species (e.g., *Funk, Caminer & Ron, 2012*; *Jungfer et al., 2013*; *Fouquet et al., 2014*; *Caminer & Ron, 2014*; *Gehara et al., 2014*; *Ferrão et al., 2016*). In recent years it has become evident that widespread species in fact represent species complexes characterized by many deeply divergent lineages, e.g., *Adenomera andreae*, *Dendropsophus minutus*, *Rhinella margaritifera*, *Scinax ruber*, *Pristimantis ockendeni*, *Pristimantis fenestratus*, *Engystomops petersi*, *Boana fasciata*, *Physalaemus petersii*, *Leptodactylus marmoratus* and *Osteocephalus taurinus* (*Fouquet et al., 2007a*; *Fouquet et al., 2007b*; *Padial & De La Riva, 2009*; *Angulo & Icochea, 2010*; *Funk, Caminer & Ron, 2012*; *Jungfer et al., 2013*; *Caminer & Ron, 2014*; *Fouquet et al., 2014*; *Gehara et al., 2014*; *Lourenço et al., 2015*). These discoveries imply that public data deposited in, for example GenBank, Gbif or IUCN, are often flawed and that the numerous metaanalyses (*Godinho & Da Silva, 2018*) based on such data may be imprecise or even inaccurate. As a consequence of not recognizing true taxonomic diversity of anurans, macroecological studies will fail to recognize actual patterns of geographic structuring, and ultimately will not contribute to our understanding of the evolutionary and ecological processes that lead to and are maintaining this diversity.

Our results suggest that the genus harbors more than twice as many species as current estimates. In the last several years the systematics and taxonomy of the genus *Amazophrynella* has begun to be elucidated (*Ávila et al., 2012*; *Rojas et al., 2014*; *Rojas et al., 2015*; *Rojas et al., 2016*). Resulting from these studies, five new species (*A. vote*, *A. manaos*, *A. amazonicola*, *A. matses* and *A. javierbustamantei*–previously mistaken for *A. minuta*) were described. With the description of the four new species in this study, the total number of nominal species reaches 11 (Fig. 25), representing an important increase in species diversity of the genus. The number of undescribed species as a percentage of total is concordant with estimates from previous studies aiming to elucidate the species diversity of Amazonian frogs (e.g., *Elmer, Dávila & Lougheed, 2007*; *Fouquet et al., 2007a*; *Fouquet et al., 2007b*; *Padial et al., 2012*; *Ron et al., 2012*; *Caminer & Ron, 2014*; *Gehara et al., 2014*; *Ferrão et al., 2016*). Therefore, our study adds to this growing body of studies, and confirms the hypothesis that the species diversity within *Amazophrynella* is much higher than currently

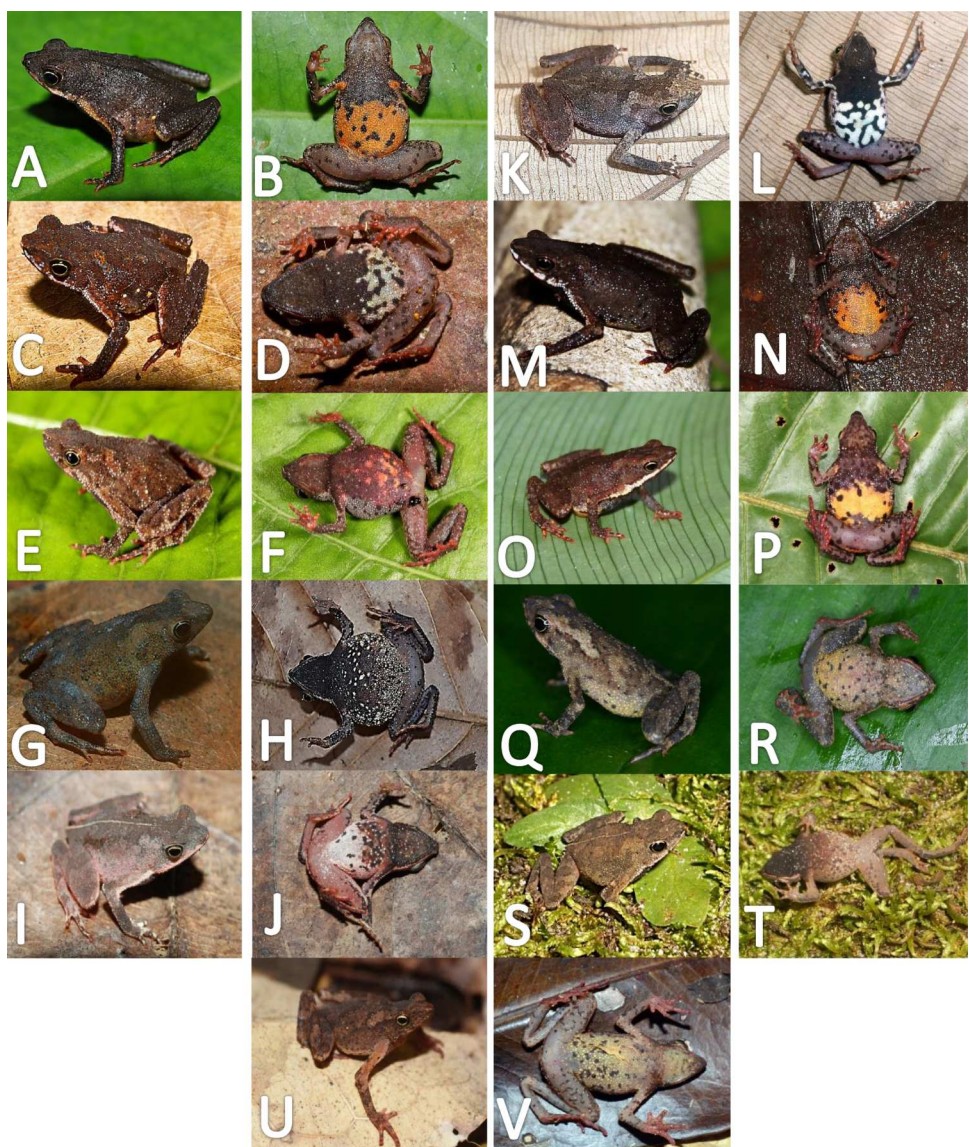

**Figure 25** **Confirmed candidate species (CCS) of *Amazophrynella*.** (A–B) *A. minuta* photo by Rommel R. Rojas; (C–D) *A. teko* sp. nov. photo by Antoine Fouquet; (E–F) *A. siona* sp. nov. photo by Santiago R. Ron; (G–H) *A. xinguensis* sp. nov. photo by Emil Hernándes-Ruz; (I–J) *A. bokermanni* photo by Marcelo Gordo; (K–L) *A. manaos* photo by Rommel R. Rojas. (M–N) *A. amazonicola* photo by Rommel R. Rojas. (O–P) *A. matses* photo by Rommel R. Rojas; (Q–R) *A. javierbustamantei* photo by Juan Carlos Chapparro; (S–T) *A. vote* photo by Robson W. Ávila; (U–V) *A. moisesii* sp. nov. photo by Paulo R. Melo-Sampaio.

accepted. The four CCS described in our study present clear differences in diagnostic morphological characters, divergence at ecological requirements and large genetic distance when compared with their sister taxa. But it should also be clear that our taxonomic decisions were conservative, and that numerous putative lineages within *Amazophrynella* still await formal description. This conservative approach aims to promote taxonomic

stability, but as a consequence continues, albeit to a lesser degree, to underestimate the true species diversity of Amazonian anurofauna.

A limiting factor of our study was the use of a single molecular marker (16S, 12S and COI mtDNA loci). The potential limitations for species delimitation using mtDNA have been discussed in literature (e.g., *Rannala & Zang, 2003*; *Yang & Rannala, 2010*; *Dupuis, Roe & Sperling, 2012*; *Fujita et al., 2012*). The use of additional nuclear markers is generally recommended as the use of these unliked markers has the potential to improve the accuracy of phylogenetic reconstructions and species delimitation. In spite of having used only mtDNA loci, our study also provides an extensive new morphological dataset, bioacoustic data and accurate collecting locality information which allowed us to associate environmental data with each specimen. All these additional data support and reinforce the inference based on the mitochondrial genes.

Our phylogenetic analysis also reveals a striking biogeographic pattern with a basal eastern and western divergence followed by a northern and southern split within both eastern/western clades (Fig. 2). Our basal east–west pattern dated to the Miocene and match similar patterns and divergence times detected in other groups of frogs (*Symula, Schulte & Summers, 2003*; *Noonan & Wray, 2006*; *Funk et al., 2007*; *Garda & Cannatella, 2007*; *Fouquet et al., 2014*). Paleoenvironmental reconstructions of Amazonian history suggest that there was a large lacustrine region in western Amazon which began to form at the beginning of the Miocene (∼24 Ma) (*Hoorn et al., 2010*). This lake and marshland system, known as Lake Pebas, existed in southwestern Amazonia, and was drained first to the north and then to the east (*Hoorn et al., 2010*). Paleoenviromental data suggest marine incursions into western Amazon during the Miocene, and *Noonan & Wray (2006)*, for example, suggest the importance of these incursions for the diversification of Amazonian anurofauna. In general, however, marine incursions remain largely untested as a diversifying force (*Noonan & Wray, 2006*; *Garda & Cannatella, 2007*; *Antonelli et al., 2010*). In addition, it is reported that in early Miocene, the Purus arch was still active, and was a prominent landscape feature in central Amazon (*Wesselingh & Salo, 2006*; *Figueiredo et al., 2009*; *Caputo & Soares, 2016*) thus this geological formation also could explain the east–west pattern as well. While other hypotheses, such as Pleistocene refugia have also been proposed to explain this east–west pattern of diversity (*Pellegrino et al., 2011*), the Miocene marine incursions have the best temporal concordance with the basal east–west divergence pattern observed in *Amazophrynella* and other Amazonian anuran groups.

The northern and southern split within both the eastern and western clades occurred in early Miocene (∼20.1 Ma) in the eastern Amazonia clades, while the diversification of the western Amazonian clade commenced in the Middle Miocene (∼16.5 Ma). The beginning of the diversification of these clades appears to be asynchronous and therefore is unlikely attributable to a single event. The more recent date of diversification of the western clade is likely to have followed the last marine incursion, i.e., a colonization of newly available habitat in western Amazon from eastern Amazon. Independent of the absolute timing these divergence events, the four subclades are restricted to north and south of the Amazon River, a common pattern in many vertebrates species groups analyzed at the Amazonia-wide scale (e.g., *Kaefer et al., 2012*; *Ribas et al., 2012*; *Fouquet et al., 2015*; *Oliveira, Carvalho & Hrbek,*

*2016*). In the case of *Amazophrynella* species, ecological characteristics such as small body side, being a *terra firme* species and being restricted to reproducing in puddles (*Rojas et al., 2016*), clearly evidences these species' inability to disperse across rivers. This in turn implies that major Amazonian rivers should limit the distributions of lineages of *Amazophrynella*, a pattern observed in our phylogeny. However, the role of rivers in driving diversification of Neotropical frogs remains controversial (see *Vences & Wake, 2007* vs. *Lougheed et al., 1999*). But it is clear that geological and climatic changes in the Miocene and Pliocene played an important role in the diversification of Amazonian vertebrates (*Bush, 1994*; *Glor, Vitt & Larson, 2001*; *Da Silva & Patton, 1998*; *Symula, Schulte & Summers, 2003*; *Santos et al., 2009*; *Kaefer et al., 2012*; *Fouquet et al., 2014*; *Gehara et al., 2014*). However, only future process-based studies and biogeographic hypotheses testing will allowed us to reveal the mechanisms (e.g., dispersion, vicariance, founder event) by which *Amazophrynella* diversified.

## ACKNOWLEDGEMENTS

We would like to thank Ariane Silva, from the herpetological collection of the Instituto Nacional de Pesquisas da Amazônia (INPA) in Manaus, Brazil, to Fernando Ayala from the Pontificia Universidad Católica de Ecuador (PUC(E) in Quito-Ecuador, and to O Aguilar and R Orellana (MUBI) for providing administrative support and part of the material for this study. RR Rojas thanks Alexander Almeida, Ian Pool Medina, Richard Naranjito Curto for field support and Mario Nunes for laboratory assistance.

### Funding

This work was supported by Coordenação de Aperfeiçoamento de Pessoal de Nivel Superior (CAPES), by the Program Incentivo para la Publicación Efectiva de Artículos Científicos en Revistas Indexadas 2016 −1° corte, (RDE 036, 20.04.2016) del Consejo Nacional de Ciencia, Tecnología e Innovación Tecnológica de Perú (CONCYTEC −FONDECYT (Cienciactiva)), by Concelho Nacional de Pesquisa (CNPq/SISBIOTA-BioPHAM grant no. CNPq 563348/2010), and an Investissement d'Avenir grant managed by Agence Nationale de la Recherche (CEBA, ref.ANR-10-LABX-25-01), France. The funders had no role in study design, data collection and analysis, decision to publish, or preparation of the manuscript.

### Grant Disclosures

The following grant information was disclosed by the authors:
Coordenação de Aperfeiçoamento de Pessoal de Nivel Superior (CAPES).
Program Incentivo para la Publicación Efectiva de Artículos Científicos en Revistas Indexadas 2016 −1° corte.
del Consejo Nacional de Ciencia: RDE 036, 20.04.2016.
Tecnología e Innovación Tecnológica de Perú (CONCYTEC −FONDECYT (Cienciactiva)).

Concelho Nacional de Pesquisa (CNPq/SISBIOTA-BioPHAM): CNPq 563348/2010. Agence Nationale de la Recherche: CEBA, ref.ANR-10-LABX-25-01.

## Competing Interests

Tomas Hrbek is an Academic Editor for PeerJ. All other authors declare that there are no competing interests.

## Author Contributions

- Rommel R. Rojas conceived and designed the experiments, performed the experiments, analyzed the data, contributed reagents/materials/analysis tools, prepared figures and/or tables, authored or reviewed drafts of the paper, approved the final draft, conducted fieldwork.
- Antoine Fouquet, Santiago R. Ron, and Juan C. Chaparro contributed reagents/materials/analysis tools, authored or reviewed drafts of the paper, approved the final draft, conducted fieldwork.
- Emil José Hernández-Ruz, Paulo R. Melo-Sampaio, Vinicius Tadeu de Carvalho, Leandra Cardoso Pinheiro, and Robson W. Avila contributed reagents/materials/analysis tools, approved the final draft, conducted fieldwork.
- Richard C. Vogt and Izeni Pires Farias contributed reagents/materials/analysis tools, authored or reviewed drafts of the paper, approved the final draft.
- Marcelo Gordo conceived and designed the experiments, contributed reagents/materials/analysis tools, authored or reviewed drafts of the paper, approved the final draft, conducted fieldwork.
- Tomas Hrbek conceived and designed the experiments, analyzed the data, contributed reagents/materials/analysis tools, authored or reviewed drafts of the paper, approved the final draft, conducted fieldwork.

## Animal Ethics

The following information was supplied relating to ethical approvals (i.e., approving body and any reference numbers):

This study involved no animal experimentation or maintenance of live animals in captivity or in the laboratory. In these instances our institution (UFAM) does not require IACUC approval. ICMBio/IBAMA, field collection permits are conditional that collection of organisms be undertaken in accordance with the ethical recommendations of the Conselho Federal de Biologia (CFBio; Federal Council of Biologists), Resolution 301 (December 8, 2012). Access to genetic resources was authorised by permit No. 034/2005/IBAMA.

## Field Study Permissions

The following information was supplied relating to field study approvals (i.e., approving body and any reference numbers):

All samples were collected with the permission of relevant country organs. Anurans were collected under the licenses ICMBio/IBAMA 41180-2 and 39792-1 of Tomas Hrbek

and Rommel R. Rojas (in Brazil), MINAN 0320 of Rommel R. Rojas (in Peru), and 001-1-IC-FAU-DNB/MA of Santiago R. Ron. In French Guiana, species of Amazophrynella are not protected and sampling does not require a license.

## DNA Deposition

The following information was supplied regarding the deposition of DNA sequences:

The sequences are available as a Supplemental File.

## Data Availability

GitHub: https://github.com/legalLab/publications.

## New Species Registration

The following information was supplied regarding the registration of a newly described species:

Publication LSID: urn:lsid:zoobank.org:pub:1C6046BE-CFC4-4060-A1CA-0C9C9C1C7A0A;

Amazophrynella Genus: urn:lsid:zoobank.org:act:04379B5D-9220-4A38-B5F1-ABD0D81D65B4;

*Amazophrynella siona* sp. nov.: urn:lsid:zoobank.org:act:66224D58-8DE0-4D5B-950D-1206FFA4AC11;

*Amazophrynella teko* sp. nov.: urn:lsid:zoobank.org:act:590F41D2-7138-42F8-8509-448602C2D040;

*Amazophrynella moisesii* sp. nov.: urn:lsid:zoobank.org:act:9984F3CB-9416-482D-8F63-5D78C8CDC032;

*Amazophrynella xinguensis* sp. nov.: urn:lsid:zoobank.org:act:55CD4C19-9A39-4DEB-BA6C-F02F9735BB77.

## Supplemental Information

Supplemental information for this article can be found online at http://dx.doi.org/10.7717/peerj.4941#supplemental-information.

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
