# Peer review of "A Pan-Amazonian species delimitation: high species diversity within the genus Amazophrynella (Anura: Bufonidae)"

_PeerJ, doi:10.7717/peerj.4941_

## Round 0.1 · original submission · Major Revisions

I agree with all three reviewers in that this manuscript represents an important contribution to the knowledge of an amazonian groups of toads. Given the scope of the paper, the results presented will be relevant for a wider audience, so the PeerJ journal is very appropriate for its publication.

However, before we can accept this paper for publication, the authors need to pay close attention to the remarks made by the reviewers which I find very appropriate and will only improve the quality of the manuscript.

Reviewer 1 ·

Basic reporting

- The article is clear, and is written in a Clear and unambiguous, professional English used throughout.
R: No comments

- Literature references, sufficient field background/context provided
R: In general, the ms offers a good background on the main subject, the taxonomy and systematics of Amazophrynella. However, the ms failed in provide introduction of one aspect investigated, the relationship between environmental variables and the taxonomy and distribution of putative species proposed.

- Professional article structure, figs, tables. Raw data shared
R: No comments

- Self-contained with relevant results to hypotheses
R: No comments

Experimental design

-Original primary research within Aims and Scope of the journal.
R: Yes, it agrees with PeerJ scope, which is based on methodological and results soundness, and it is also a original research.

-Research question well defined, relevant & meaningful. It is stated how research fills an identified knowledge gap.
R: The research question is well defined, the taxonomy and systematics of Amazophrynella. However, as said above, the htpothesis of relationship among Amazophrynella distribution, taxonomy, and environmental variables is unclear and poorly explored. Additionally, one aspect missing on the discussion (and the intro) is how this paper advances the phylogenetic relationship and taxonomy among Amazophrynella clades. The authors need to be specific in highlight samples from new localities and new clades included only in this ms (and not in previous studies like Fouquet et al 2012, Rojas et al 2016).

-Rigorous investigation performed to a high technical & ethical standard.
R: No comments

-Methods described with sufficient detail & information to replicate
R: The methods fail in provide more details on the parameters used during tree search on Bayesian analysis, and also on dating analysis on BEAST. Please see attached comments on pdf file for more details.

Validity of the findings

-Impact and novelty not assessed. Negative/inconclusive results accepted. Meaningful replication encouraged where rationale & benefit to literature is clearly stated.
R: The ms deals with a very interesting question, the taxonomy of a cryptic species groups of frogs in one of the most biodiverse ecosystems in the world. There is no replication of methods, however, the authors must be explicit in how many samples are new and not included in previous studies dealing with Amazophrynella systematics and taxonomy.

-Data is robust, statistically sound, & controlled.
R: Yes, however, all the conclusions are based only in mtDNA molecular markers. Although this is not an impediment for the authors conclusions (at least from my point of view), is well know on the literature (see pdf file for references) the limitations of single marker results for taxonomy, species delimitation and phylogenetic relationships conclusions. I suggest to add a paragraph acknowledging the limitations of results.

-Conclusion are well stated, linked to original research question & limited to supporting results.
R: In general it is, although the relationship among taxonomy, distribution and environmental variables was not explored on the discussion. Also, the ms lacks a final session of conclusion, summarizing the advances, limitations and future prospects.

-Speculation is welcome, but should be identified as such.
R: On this, maybe the biogeographic portion of discussion should be indicated as speculation, as no formal test was applied on this (e.g. BioGeoBears, for example).

Additional comments

The ms deals with the biodiversity of a very interesting group, in which was considered, until very recently, as a single, widespread species. By disentangle the diversity and systematics of lineages and species, as well as to provide a reproducible taxonomic framework on Amazophrynella, I considered this ms as very important and relevant. However, I am not satisfied with methodological and results reporting (refers to pdf file for more details) and discussion, specially the environmental variables, which appears pointless to include, and is making the ms more confusing. I detect some possible errors in morphological PCA analysis and in the geological epochs bar on figure 2. Additionally, in the pdf file I highlight several small issues that the authors should deal with. For that reasons, I recommended major revisions. I also put myself available for future revisions.

Annotated reviews are not available for download in order to protect the identity of reviewers who chose to remain anonymous.

Reviewer 2 ·

Basic reporting

Dear editor,

I have read the manuscript entitled “A Pan-Amazonian species delimitation: high species diversity within the genus Amazophrynella (Anura:Bufonidae)“ submitted to PeerJ for consideration for publication.

The study represents a robust systematics revision of the toads genus Amazophrynella, with taxonomic implications. The results presented on it will eventually be relevant for a wide audience interested on species diversity and biogeographical patterns of Amazonian anurofauna.

The article is well structured, provide sufficient background on its field, abundant analyses and raw data.

Experimental design

The approach used is highly integrative, and the methods well defined in terms of species concept and delimitation, quantitative analysis implemented, and detailed descriptions.

Validity of the findings

Although the authors assumed a conservative criteria for their taxonomic decisions, the advances are inestimable for future studies on the group.

Additional comments

I have some punctual observations for the authors:

Line 131. I suppose that “heap shape” means “head shape”.
Line 159. Please, exchange “GRT” with GTR.
Line 203. The authors stated that the “effects of variables on percent variance explained” was evaluated. But, this may seem confusing because PCA is an exploratory approach, not a statistical test. In fact, it detects the individual contribution of each variable for total variance. It’s better to rewrite this sentence.
Line 204. Please, mention if DFA was also made on the size-free dataset.
Line 208. Please, mention here that univariate test was made for each morphometric variable separately.
Line 240. The PTP model recovered eighteen lineages, not seventeen.
Line 260. It would be interesting to state how many of the lineages are represented on quantitative analysis.
Line 266. A. bokermanni appears on the graphics but are not presented on the morphometric dataset (table S6).
Line 266. Eigenvalues are missing on Appendix S7.
Line 271. I suggest to the authors to take of the grey background of DFA graphics (Fig. 3).
Line 284. Same as line 260.
Line 303. …environmental variables that most contributed…
Line 899. …by Conselho Nacional de Pesquisa…

Figure 1. As node support represents Posterior Probabilities, it should be presented as fractional numbers between 0-1, or it must be clearly stated on captions that it is presented as percentage (pp *100).
Figure 2. On Fig. 2 and Fig. S5, Dendrophryniscus proboscideus is wrongly referred as D. boulengeri.

Reviewer 3 ·

Basic reporting

All good. Please refer to my comments to the authors, below.

Experimental design

All good. Please refer to my comments to the authors, below.

Validity of the findings

All good. Please refer to my comments to the authors, below.

Additional comments

This is significant and important contribution to our knowledge of taxonomic diversity in a widespread yet poorly known group of Amazonian toads, the genus Amazophrynella. The scope of the study is impressive, and the data and methods are appropriate for the study's goals.

Overall, my comments refer to minor clarifications and suggestions about grammar and style. However, one issue may deserve further explanation and even reanalysis - the implementation of a species-tree method (based on the multi-species coalescent) on a dataset that effectively corresponds to only one (mitochondrial) locus. Please refer to my comments to the authors below.

1. Line 67. You state that "delimiting species solely based on molecular characters or genetic distances harbors potential pitfalls that have been extensively documented". I think that providing a few examples of these problems may improve the justification of your integrative approach, and hopefully inspire similar comprehensive studies.

2. Line 109. Please replace "corresponds" with "correspond".

3. Line 112. Please replace "was" with "were".

4. Line 114. It is not clear why you cite Sabaj (2016) as a reference for museum acronyms given that all museum acronyms are spelled out in the following paragraph.

5. Line 132. "venter coloration" is listed twice.

6. Line 129. The qualitative and quantitative morphological traits used are described in distinct sections of the text, with the molecular methods in-between, which feels a bit awkward. It may be best to have the morphological traits presented in the same section (or at least consecutively).

7. Line 133. Did you use a ventral incision to perform gonadal analyses? Please clarify.

8. Line 159. Did you use gene partitions? Alternatively, did you implement a single model of evolution across all genes? Please clarify.

9. Line 163. A brief description of the Poisson tree process (PTP) model would be welcome for those not familiar with this method, given it is not necessarily standard in taxonomic studies (as compared, for instance, to the now standard use of phylogenetic trees).

10. Line 171. Please replace "generated" with "generate".

11. Line 171. Please replace "trees" with "tree".

12. Line 171. By "species tree in Beast", do you mean using the Star Beast method based on the multi-species coalescent? If so, did you link all your partitions into a single gene tree? Please note that the mitochondrial genes that you used effectively correspond to a single locus and should be treated as so. Otherwise, Star Beast has treated each gene as an independent locus, which would be a violation of model assumptions.

13. Also, please note that if you only used one (mitochondrial) locus, Star Beast is not the right analysis to do. If you have a single locus, a mitochondrial gene tree (MrBayes) is the best approximation of the species tree that you will get.

14. Line 172. Are you using normal distributions for these calibration priors? Please clarify.

15. Line 180. By "consensus tree", do you mean a maximum-clade credibility tree as summarized in Tree Annotator? Please clarify.

16. Line 197. What kind of "multivariate analyses" were used? What programs were used for that? If this refers to something that is described in detail later in the text, please make this fact clear.

17. Line 201. From this description, is not clear why you separated the eastern and western clades in PCA analyses. Please justify.

18. Line 207. Please replace "incorrectly" with "incorrect".

19. Line 268. It is unclear what is meant by "The PCA of the eastern and western clades revealed a grouping of specimens in the morphometric space".

20. Line 291. As above, it is unclear what is meant by "the PCA of the eastern and western clades revealed a grouping of specimens in the multivariate space".

21. The descriptions (for instance, of the holotypes) are not entirely consistent across the four new species. Different traits are often presented, and overlapping traits are often presented in different orders. Please be consistent.

22. Line 350, 361 and throughout. Please replace "creamy" with "cream", "creamish" or "cream-colored".

23. Line 351. Please include comparisons between A. teko sp. nov. and the other newly described taxa as well (only comparisons with previously described Amazophrynella species are provided).

24. Line 356 and throughout. Some of the trait comparisons between species rely on differences of the order of 0.01 mm, which is hardly useful (and even reliable).

25. Line 356 and throughout. Ratios between measurements expressed in the same unit are unit-less (i.e., TAL/SVL is not expressed in mm).

26. Line 354. Please replace "minor" with "smaller".

27. Line 355. Please replace "minor" with "smaller".

28. Line 402. Coloration in life? Please clarify.

29. Line 408 and elsewhere. Please remove "was".

30. Line 431. Please replace "need" with "needs".

31. Line 488. Please replace "Noviembre" with "November".

32. Line 502 and elsewhere. By "fingers", do you mean "fringes"?

33. Line 503. Here again, some species comparisons are missing.

34. Line 531, 766, and elsewhere. By "vocal", do you mean "vocal sac"?

35. Line 545. I'm not sure referring to a website in this way is appropriate.

36. Line 577. "opening directed" sounds odd. Please verify.

37. Line 580. A dot (".") is missing before "Oral".

38. Line 644. Please replace "largely" with "large".

39. Line 654. Is "black dots on venter (black dots)" correct? So, there is no difference between the two species in this trait?

40. Line 656. Please replace "have" with "having".

41. Line 656, 657. Who is "their" referring to - the new species or the other species? Please clarify.

42. Line 670, 675. State of cloacal opening shows up twice in this paragraph.

43. Line 682. "The abundance of granules on ventral surfaces vary in abundance" - please fix the redundancy statements.

44. Line 685. Please replace "between" with "from".

45. Line 753. Please replace "For" with "From".

46. Line 859. Please replace "marchland" with "marshland".

47. Line 1195, 1196. Please be consistent with legend terminology to avoid confusion - you use "dots" and "circles" to refer to the same thing, then "dotted" to refer to something else, and then "dotted and colored circles".

48. Line 1199. Please replace "calibrate" with "calibrated".

49. Line 1228, 1230. Please replace "variations" with "variation".

50. Figs of specimens: Some figures have values corresponding to the scale bars indicated in the legend, others have it on the image. Moreover, different bars correspond to different scales. Please make scaling scheme (and legend) consistent.

51. Fig. 1. The colors of different clades are too similar of overlap - please consider combining colors with symbols (or implementing some other scheme) to facilitate telling them apart on maps.

52. Fig. 5, 21, and 22. Are you including data of specimens from clades identified by "aff." along with data from closely-related nominal species? For instance, do data for A. vote in this table include also data from A. aff. vote sp 1 and A. aff. vote sp 2?

53. Fig. 6, 7, and 8. Please add to figure legends (within parentheses) the corresponding name/state given to each species' traits (e.g., blotches, dots, small blotches, etc), so that readers can understand your terminology with graphical examples. Same thing for hand tubercles (round, oval, etc) and snout in profile (truncated, rounded, etc), etc.

54. Table 1. In the legend, please correct "Uncorfirmed" (misspelled).

55. Table 1. Add "DCL" to the legend and spell out the acronym.

56. Table 1. Do you expect the uncategorized lineage sp. 1 to correspond to A. teko sp. nov.? They seem to be morphologically uniform... same for A. minuta and A. aff. minuta sp1. I would appreciate some comments about the taxonomic status of those clades indicated with "aff.", which may guide future studies and support biological inventories in the region.

57. Table 1. Why do you present p-distances for only part of the listed clades?

58. Table 1. Some of the traits are very subjective and the terminology is a bit confusing. How do you discriminate between "granular" and "highly granular" skin texture? What is the difference between "small dots" and "tiny dots", or between them and "medium size dots"? How are "blotches" different from "medium blotches"? Perhaps images could help here - see my comment about Fig. 6, 7, and 8, above.

59. Table 1. You present SW, NW, NE and SE clades, but most of the text refers only to east and west clades. Please be consistent.

60. Table 2 and 3. Are you including data of specimens from clades identified by "aff." along with data from closely-related nominal species? For instance, do data for A. vote in this table include also data from A. aff. vote sp 1 and A. aff. vote sp 2?

61. Table 4 and 5. Please include "aff." in the column names of corresponding clades (e.g., replace "A. vote sp. 1" to "A. aff. vote sp 1").

Congratulations for this formidable initiative!

---

## Round 0.2 · Minor Revisions

I have evaluated the revision and I am very happy with how the authors dealt with all of the comments from the reviewers. I believe the importance of including environmental variables in the analyses is now clear, and the second major point which was the limitations of using only mitochondrial markers for species delimitations was also properly addressed by the authors.

I think this paper is very interesting and it is almost ready to be accepted. I myself am not a native english speaker but found several language errors which I have highlighted in the appended pdf. I would ask the authors to have a fluent English speaker to correct the manuscript for language as it is a requirement of the journal to have a "clear, unambiguous, technically correct English".

As I said, a very interesting paper that contributes quite a bit to the knowledge of tropical biodiversity.

---

## Round 0.3 · accepted · Accept

I think this paper is ready to be accepted for publication. The authors did a good job improving the language which was the only observation made to the last version.

#